# Counterfactual Data Augmentation using Locally Factored Dynamics

**Silviu Pitis, Elliot Creager, Animesh Garg**
Department of Computer Science, University of Toronto, Vector Institute
{spitis, creager, garg}@cs.toronto.edu

## Abstract

Many dynamic processes, including common scenarios in robotic control and reinforcement learning (RL), involve a set of interacting subprocesses. Though the subprocesses are not independent, their interactions are often sparse, and the dynamics at any given time step can often be decomposed into *locally independent* causal mechanisms. Such local causal structures can be leveraged to improve the sample efficiency of sequence prediction and off-policy reinforcement learning. We formalize this by introducing local causal models (LCMs), which are induced from a global causal model by conditioning on a subset of the state space. We propose an approach to inferring these structures given an object-oriented state representation, as well as a novel algorithm for Counterfactual Data Augmentation (CoDA). CoDA uses local structures and an experience replay to generate counterfactual experiences that are causally valid in the global model. We find that CoDA significantly improves the performance of RL agents in locally factored tasks, including the batch-constrained and goal-conditioned settings.[1]

## 1 Introduction

High-dimensional dynamical systems are often composed of simple subprocesses that affect one another through sparse interaction. If the subprocesses *never* interacted, an agent could realize significant gains in sample efficiency by globally factoring the dynamics and modeling each subprocess independently [28, 29]. In most cases, however, the subprocesses do *eventually* interact and so the prevailing approach is to model the entire process using a monolithic, unfactored model. In this paper, we take advantage of the observation that *locally—during the time between their interactions—the subprocesses are causally independent*. By locally factoring dynamic processes in this way, we are able to capture the benefits of factorization even when their subprocesses interact on the global scale.

Consider a game of billiards, where each ball can be viewed as a separate physical subprocess. Predicting the opening break is difficult because all balls are mechanically coupled by their initial placement. Indeed, a dynamics model with dense coupling amongst balls may seem sensible when considering the expected outcomes over the course of the game, as each ball has a non-zero chance of colliding with the others. But at any given timestep, interactions between balls are usually sparse.

One way to take advantage of sparse interactions between otherwise disentangled entities is to use a structured state representation together with a graph neural network or other message passing transition model that captures the local interactions [26, 39]. When it is tractable to do so, such architectures can be used to model the world dynamics directly, producing transferable, task-agnostic models. In many cases, however, the underlying processes are difficult to model precisely, and model-free [46, 87] or task-oriented model-based [18, 63] approaches are less biased and exhibit superior performance. In this paper we argue that *knowledge of whether or not local interactions*

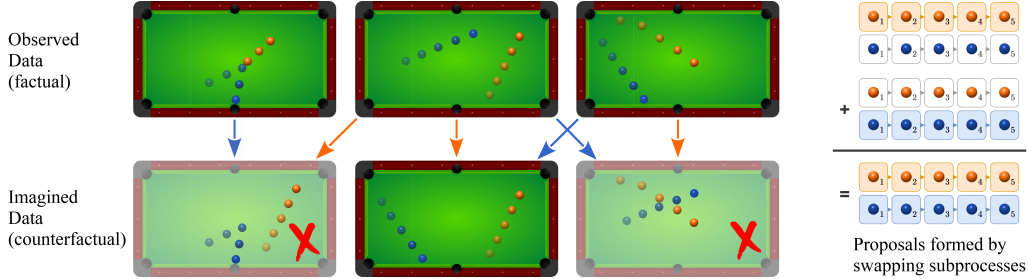

Figure 1: **Counterfactual Data Augmentation (CoDA)**. Given 3 factual samples, knowledge of the local causal structure lets us mix and match factored subprocesses to form counterfactual samples. The first proposal is rejected because one of its factual sources (the blue ball) is not locally factored. The third proposal is rejected because it is not itself factored. The second proposal is accepted, and can be used as additional training data for a reinforcement learning agent.

*occur is useful in and of itself*, and can be used to generate causally-valid counterfactual data even in absence of a forward dynamics model. In fact, if two trajectories have the same local factorization in their transition dynamics, then under mild conditions we can produce new counterfactually plausible data using our proposed **Counterfactual Data Augmentation (CoDA)** technique, wherein factorized subspaces of observed trajectory pairs are swapped (Figure 1). This lets us sample from a counterfactual data distribution by stitching together subsamples from observed transitions. Since CoDA acts only on the agent's training data, it is compatible with any agent architecture (including unfactored ones).

In the remainder of this paper, we formalize this data augmentation strategy and discuss how it can improve performance of model-free RL agents in locally factored tasks. Our main contributions are:

1. We define local causal models (LCMs), which are induced from a global model by conditioning on a subset of the state space, and show how local structure can simplify counterfactual reasoning.

2. We introduce CoDA as a generalized data augmentation strategy that is able to leverage local factorizations to manufacture unseen, yet causally valid, samples of the environment dynamics. We show that goal relabeling [36, 1] and visual augmentation [2, 46] are instances of CoDA that use global independence relations and we propose a locally conditioned variant of CoDA that swaps independent subprocesses to form counterfactual experiences (Figure 1).

3. Using an attention-based method for discovering local causal structure in a disentangled state space, we show that our CoDA algorithm significantly improves the sample efficiency in standard, batch-constrained, and goal-conditioned reinforcement learning settings.

## 2 Local Causality in MDPs

### 2.1 Preliminaries and Problem Setup

The basic model for decision making in a controlled dynamic process is a Markov Decision Process (MDP), described by tuple $\langle \mathcal{S}, \mathcal{A}, P, R, \gamma \rangle$ consisting of the state space, action space, transition function, reward function, and discount factor, respectively [72, 83]. Note that MDPs generalize uncontrolled Markov processes (set $A = \emptyset$), so that our work applies also to sequential prediction. We denote individual states and actions using lowercase $s \in \mathcal{S}$ and $a \in \mathcal{A}$, and variables using the uppercase $S$ and $A$ (e.g., $s \in \text{range}(S) \subseteq \mathcal{S}$). A policy $\pi : \mathcal{S} \times \mathcal{A} \to [0, 1]$ defines a probability distribution over the agent's actions at each state, and an agent is typically tasked with learning a parameterized policy $\pi_\theta$ that maximizes value $\mathbb{E}_{P,\pi} \sum_t \gamma^t R(s_t, a_t)$.

In most non-trivial cases, the state $s \in \mathcal{S}$ can be described as an object hierarchy together with global context. For instance, this decomposition will emerge naturally in any simulated process or game that is defined using a high-level programming language (e.g., the commonly used Atari [7] or Minecraft [35] simulators). In this paper we consider MDPs with a single, known top-level decomposition of the state space $\mathcal{S} = \mathcal{S}^1 \oplus \mathcal{S}^2 \oplus \cdots \oplus \mathcal{S}^n$ for fixed $n$, leaving extensions to hierarchical decomposition and

multiple representations [32, 17], dynamic factor count $n$ [92], and (learned) latent representations [13] to future work. The action space might be similarly decomposed: $\mathcal{A} = \mathcal{A}^1 \oplus \mathcal{A}^2 \oplus \cdots \oplus \mathcal{A}^m$.

Given such state and action decompositions, we may model time slice $(t, t+1)$ using a structural causal model (SCM) $\mathcal{M}_t = \langle V_t, U_t, \mathcal{F} \rangle$ ([65], Ch. 7) with directed acyclic graph (DAG) $\mathcal{G}$, where:

- $V_t = \{V^i_{t[+1]}\}^{2n+m}_{i=0} = \{S^1_t \ldots S^n_t, A^1_t \ldots A^m_t, S^1_{t+1} \ldots S^n_{t+1}\}$ are the nodes (variables) of $\mathcal{G}$.

- $U_t = \{U^i_{t[+1]}\}^{2n+m}_{i=0}$ is a set of noise variables, one for each $V^i$, determined by the initial state, past actions, and environment stochasticity. We assume that noise variables at time $t+1$ are independent from other noise variables: $U^i_{t+1} \perp\!\!\!\perp U^j_{t[+1]} \forall i, j$. The instance $u = (u^1, u^2, \ldots, u^{2n+m})$ of $U_t$ denotes an individual realization of the noise variables.

- $\mathcal{F} = \{f^i\}^{2n+m}_{i=0}$ is a set of functions ("structural equations") that map from $U^i_{t[+1]} \times \mathrm{Pa}(V^i_{t[+1]})$ to $V^i_{t[+1]}$, where $\mathrm{Pa}(V^i_{t[+1]}) \subset V_t \setminus V^i_{t[+1]}$ are the parents of $V^i_{t[+1]}$ in $\mathcal{G}$; hence each $f^i$ is associated with the set of incoming edges to node $V^i_{t[+1]}$ in $\mathcal{G}$; see, e.g., Figure 2 (center).

Note that while $V_t$, $U_t$, and $\mathcal{M}_t$ are indexed by $t$ (their distributions change over time), the structural equations $f^i \in \mathcal{F}$ and causal graph $\mathcal{G}$ represent the global transition function $P$ and apply at all times $t$. To reduce clutter, we drop the subscript $t$ on $V$, $U$, and $\mathcal{M}$ when no confusion can arise.

Critically, we require the set of edges in $\mathcal{G}$ (and thus the number of inputs to each $f^i$) to be *structurally minimal* ([67], Remark 6.6).

**Assumption** (Structural Minimality). *$V^j \in Pa(V^i)$ if and only if there exists some $\{u^i, v^{-ij}\}$ with $u^i \in range(U^i), v^{-ij} \in range(V \setminus \{V^i, V^j\})$ and pair $(v^j_1, v^j_2)$ with $v^j_1, v^j_2 \in range(V^j)$ such that $v^i_1 = f^i(\{u^i, v^{-ij}, v^j_1\}) \neq f^i(\{u^i, v^{-ij}, v^j_2\}) = v^i_2$.*

Intuitively, structural minimality says that $V^j$ is a parent of $V^i$ if and only if setting the value of $V^j$ can have a nonzero *direct* effect[2] on the child $V^i$ through the structural equation $f^i$. The structurally minimal representation is unique [67].

Given structural minimality, we can think of edges in $\mathcal{G}$ as representing global causal dependence. The probability distribution of $S^i_{t+1}$ is fully specified by its parents $\mathrm{Pa}(S^i_{t+1})$ together with its noise variable $U_i$; that is, we have $P(S^i_{t+1} \mid S_t, A_t) = P(S^i_{t+1} \mid \mathrm{Pa}(S^i_{t+1}))$ so that $S^i_{t+1} \perp\!\!\!\perp V^j \mid \mathrm{Pa}(S^i_{t+1})$ for all nodes $V^j \notin \mathrm{Pa}(S^i_{t+1})$. We call an MDP with this structure a *factored MDP* [37]. When edges in $\mathcal{G}$ are sparse, factored MDPs admit more efficient solutions than unfactored MDPs [28].

## 2.2 Local Causal Models (LCMs)

**Limitations of Global Models** Unfortunately, even if states and actions can be cleanly decomposed into several nodes, in most practical scenarios the DAG $\mathcal{G}$ is fully connected (or nearly so): since the $f^i$ apply globally, so too does structural minimality, and edge $(S^i_k, S^j_{k+1})$ at time $k$ is present so long as there is a single instance—at any time $t$, no matter how unlikely—in which $S^i_t$ influences $S^j_{t+1}$. In the words of Andrew Gelman, "*there are (almost) no true zeros*" [24]. As a result, the factorized causal model $\mathcal{M}_t$, based on globally factorized dynamics, rarely offers an advantage over a simpler causal model that treats states and actions as monolithic entities (e.g., [12]).

**LCMs** Our key insight is that for each pair of nodes $(V^i_t, S^j_{t+1})$ with $V^i_t \in \mathrm{Pa}(S^j_{t+1})$ in $\mathcal{G}$, there often exists a large subspace $\mathcal{L}^{(j \perp i)} \subset \mathcal{S} \times \mathcal{A}$ for which $S^j_{t+1} \perp\!\!\!\perp V^i_t \mid \mathrm{Pa}(S^j_{t+1}) \setminus V^i_t, (s_t, a_t) \in \mathcal{L}^{(j \perp i)}$. For example, in case of a two-armed robot (Figure 2), there is a large subspace of states in which the two arms are too far apart to influence each other physically. Thus, if we restrict our attention to $(s_t, a_t) \in \mathcal{L}^{(j \perp i)}$, we can consider a *local* causal model $\mathcal{M}_t^{\mathcal{L}^{(j \perp i)}}$ whose local DAG $\mathcal{G}^{\mathcal{L}^{(j \perp i)}}$ is strictly sparser than the global DAG $\mathcal{G}$, as the structural minimality assumption applied to $\mathcal{G}^{\mathcal{L}^{(j \perp i)}}$ implies that there is no edge from $V^i_t$ to $S^j_{t+1}$. More generally, for any subspace $\mathcal{L} \subseteq S \times A$, we can induce the Local Causal Model (LCM) $\mathcal{M}_t^{\mathcal{L}} = \langle V_t^{\mathcal{L}}, U_t^{\mathcal{L}}, \mathcal{F}^{\mathcal{L}} \rangle$ with DAG $\mathcal{G}^{\mathcal{L}}$ from the global model $\mathcal{M}_t$ as:

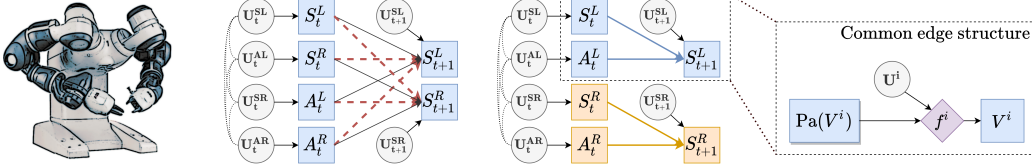

Figure 2: A two-armed robot (**left**) might be modeled as an MDP whose state and action spaces decompose into left and right subspaces: $\mathcal{S} = \mathcal{S}^L \oplus \mathcal{S}^R$, $\mathcal{A} = \mathcal{A}^L \oplus \mathcal{A}^R$. Because the arms can touch, the global causal model (**center left**) between time steps is fully connected, even though left-to-right and right-to-left connections (dashed red edges) are rarely active. By restricting our attention to the subspace of states in which left and right dynamics are independent we get a local causal model (**center right**) with two components that can be considered separately for training and inference.

- $V_t^{\mathcal{L}} = \{V_{t[+1]}^{\mathcal{L},i}\}_{i=0}^{2n+m}$, where $P(V_{t[+1]}^{\mathcal{L},i}) = P(V_{t[+1]}^i \,|\, (s_t, a_t) \in \mathcal{L})$.

- $U_t^{\mathcal{L}} = \{U_{t[+1]}^{\mathcal{L},i}\}_{i=0}^{2n+m}$, where $P(U_{t[+1]}^{\mathcal{L},i}) = P(U_{t[+1]}^i \,|\, (s_t, a_t) \in \mathcal{L})$.

- $\mathcal{F}^{\mathcal{L}} = \{f^{\mathcal{L},i}\}_{i=0}^{2n+m}$, where $f^{\mathcal{L},i} = f^i|_{\mathcal{L}}$ ($f^i$ with range of input variables restricted to $\mathcal{L}$). Due to structural minimality, the signature of $f^{\mathcal{L},i}$ may shrink (as the range of the relevant variables is now restricted to $\mathcal{L}$), and corresponding edges in $\mathcal{G}$ will not be present in $\mathcal{G}^{\mathcal{L}}$.[3]

In case of the two-armed robot, conditioning on the arms being far apart simplifies the global DAG to a local DAG with two connected components (Figure 2). This can make counterfactual reasoning considerably more efficient: given a factual situation in which the robot's arms are far apart, we can carry out separate counterfactual reasoning about each arm.

**Leveraging LCMs**  To see the efficiency therein, consider a general case with global causal model $\mathcal{M}$. To answer the counterfactual question, "*what might the transition at time $t$ have looked like if component $S_t^i$ had value $x$ instead of value $y$?*", we would ordinarily apply Pearl's do-calculus to $\mathcal{M}$ to obtain submodel $\mathcal{M}_{\text{do}(S_t^i = x)} = \langle V, U, \mathcal{F}_x \rangle$, where $\mathcal{F}_x = \mathcal{F} \setminus f^i \cup \{S_t^i = x\}$ and incoming edges to $S_t^i$ are removed from $\mathcal{G}_{\text{do}(S_t^i = x)}$ [65]. The component distributions at time $t + 1$ can be computed by reevaluating each function $f^j$ that depends on $S_t^i$. When $S_t^i$ has many children (as is often the case in the global $\mathcal{G}$), this requires one to estimate outcomes for many structural equations $\{f^j | V^j \in \text{Children}(V_t^i)\}$. But if both the original value of $S_t$ (with $S_t^i = y$) and its new value (with $S_t^i = x$) are in the set $\mathcal{L}$, the intervention is "within the bounds" of local model $\mathcal{M}^{\mathcal{L}}$ and we can instead work directly with local submodel $\mathcal{M}^{\mathcal{L}}_{\text{do}(S_t^i = x)}$ (defined accordingly). The validity of this follows from the definitions: since $f^{\mathcal{L},j} = f^j|_{\mathcal{L}}$ for all of $S_t^i$'s children, the nodes $V_t^k$ for $k \neq i$ at time $t$ are held fixed, and the noise variables at time $t + 1$ are unaffected, the distribution at time $t + 1$ is the same under both models. When $S_t^i$ has fewer children in $\mathcal{M}^{\mathcal{L}}$ than in $\mathcal{M}$, this reduces the number of structural equations that need to be considered.

## 3 Counterfactual Data Augmentation

We hypothesize that local causal models will have several applications, and potentially lead to improved agent designs, algorithms, and interpretability. In this paper we focus on improving off-policy learning in RL by exploiting causal independence in local models for **Counterfactual Data Augmentation (CoDA)**. CoDA augments real data by making counterfactual modifications to a subset of the causal factors at time $t$, leaving the rest of the factors untouched. Following the logic outlined in the Subsection 2.2, this can understood as manufacturing "fake" data samples using the counterfactual model $\mathcal{M}^{[\mathcal{L}]}_{\text{do}(S_t^{i\ldots j} = x)}$, where we modify the causal factors $S_t^{i\ldots j}$ and resample their children. While this is always possible using a model-based approach if we have good models of the structural equations, it is particularly nice when the causal mechanisms are independent, as we can do counterfactual reasoning directly by reusing subsamples from observed trajectories.

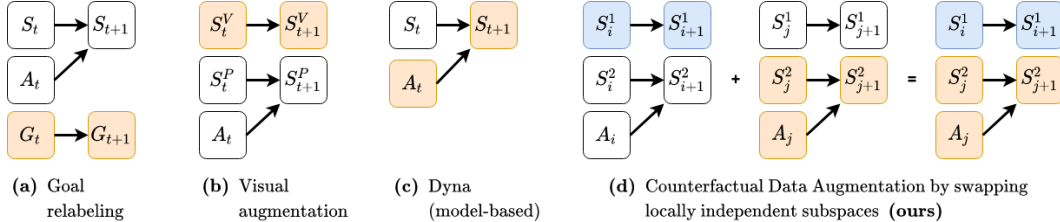

Figure 3: *Four instances of CoDA; orange nodes are relabeled, noise variables omitted for clarity.* **First:** Goal relabeling [36], including HER [1], augments transitions with counterfactual goals. **Second:** Visual feature augmentation [2, 46] uses domain knowledge to change visual features $S_t^V$ (such as textures, lighting, and camera positions) that the designer knows do not impact the physical state $S_{t+1}^P$. **Third:** Dyna [82], including MBPO [34], augments real states with new actions and resamples the next state using a learned dynamics model. **Fourth (ours):** Given two transitions that share local causal structures, we propose to swap connected components to form new transitions.

**Definition.** *The causal mechanisms represented by subgraphs $\mathcal{G}_i, \mathcal{G}_j \subset \mathcal{G}$ are **independent** when $\mathcal{G}_i$ and $\mathcal{G}_j$ are disconnected in $\mathcal{G}$.*

When $\mathcal{G}$ is divisible into two (or more) connected components, we can think of each subgraph as an independent causal mechanism that can be reasoned about separately.

Existing data augmentation techniques can be interpreted as specific instances of CoDA (Figure 3). For example, goal relabeling [36], as used in Hindsight Experience Replay (HER) [1] and Q-learning for Reward Machines [33], exploits the independence of the goal dynamics $G_t \mapsto G_{t+1}$ (identity map) and the next state dynamics $S_t \times A_t \mapsto S_{t+1}$ in order to relabel the goal variable $G_t$ with a counterfactual goal. While the goal relabeling is done model-free, we typically assume knowledge of the goal-based reward mechanism $G_t \times S_t \times A_t \times S_{t+1} \mapsto R_{t+1}$ to relabel the reward, ultimately mixing model-free and model-based reasoning. Similarly, visual feature augmentation, as used in reinforcement learning from pixels [46, 41] and sim-to-real transfer [2], exploits the independence of the physical dynamics $S_t^P \times A_t \mapsto S_{t+1}^P$ and visual feature dynamics $S_t^V \mapsto S_{t+1}^V$ such as textures and camera position, assumed to be static ($S_{t+1}^V = S_t^V$), to counterfactually augment visual features. Both goal relabeling and visual data augmentation rely on *global* independence relationships.

We propose a novel form of Counterfactual Data Augmentation that uses knowledge of *local* independence relationships. In particular, we observe that whenever an environment transition is within the bounds of some local model $\mathcal{M}^{\mathcal{L}}$ whose graph $\mathcal{G}^{\mathcal{L}}$ has the locally independent causal mechanism $\mathcal{G}_i$ as a disconnected subgraph (note: $\mathcal{G}_i$ itself need not be connected), that transition contains an unbiased sample from $\mathcal{G}_i$. Thus, given two transitions in $\mathcal{L}$, we may mix and match the samples of $\mathcal{G}_i$ to generate counterfactual data, so long as the resulting transitions are themselves in $\mathcal{L}$.

**Remark 3.1.** *How much data can we generate using our CoDA algorithm? If we have $n$ independent samples from subspace $\mathcal{L}$ whose graph $\mathcal{G}^{\mathcal{L}}$ has $m$ connected components, we have $n$ choices for each of the $m$ components, for a total of $n^m$ CoDA samples—an **exponential** increase in data! One might term this the "blessing of independent subspaces."*

**Remark 3.2.** *Our discussion has been at the level of a single transition (time slice $(t, t+1)$), which is consistent with the form of data that RL agents typically consume. But we could also use CoDA to mix and match locally independent components over several time steps (see, e.g., Figure 1).*

**Remark 3.3.** *As is typical, counterfactual reasoning changes the data distribution. While off-policy agents are typically robust to distributional shift, future work might explore different ways to control or prioritize the counterfactual data distribution [77, 43]. We note, however, that certain prioritization schemes may introduce selection bias [31], effectively entangling otherwise independent causal mechanisms (e.g., HER's "future" strategy [1] may introduce "hindsight bias" [45, 78]).*

**Remark 3.4.** *The global independence relations relied upon by goal relabeling and image augmentation are incredibly general, as evidenced by their wide applicability. We posit that certain local independence relations are similarly general. For example, the physical independence of objects separated by space (the billiards balls of Figure 1, the two-armed robot of Figure 2, and the environments used in Section 4), and the independence between an agent's actions and the truth of (but not belief about) certain facts the agent is ignorant of (e.g., an opponent's true beliefs).*

---

**Algorithm 1** Mask-based Counterfactual Data Augmentation (CoDA)

---

**function** CODA(transition t1, transition t2):
    s1, a1, s1' ← t1
    s2, a2, s2' ← t2
    m1, m2 ← MASK(s1, a1), MASK(s2, a2)
    D1 ← COMPONENTS(m1)
    D2 ← COMPONENTS(m2)
    d ← random sample from (D1 ∩ D2)
    s̃, ã, s̃' ← copy(s1, a1, s1')
    s̃[d], ã[d], s̃'[d] ← s2[d], a2[d], s2'[d]
    D̃ ← COMPONENTS(MASK(s̃, ã))
    **return** (s̃, ã, s̃') **if** d ∈ D̃ **else** ∅

**function** MASK(state s, action a):
    Returns $(n+m) \times (n)$ matrix indicating if the $n$ next state components (columns) locally depend on the $n$ state and $m$ action components (rows).

**function** COMPONENTS(mask m):
    Using the mask as the adjacency matrix for $\mathcal{G}^{\mathcal{L}}$ (with dummy columns for next action), finds the set of connected components $C = \{C_j\}$, and returns the set of independent components $D = \{\mathcal{G}_i = \bigcup_k \mathcal{C}_k^i \,|\, \mathcal{C}^i \subset \text{powerset}(C)\}$.

---

**Implementing CoDA**   We implement CoDA, as outlined above and visualized in Figure 3(d), as a function of two factual transitions and a mask function $M(s_t, a_t) : \mathcal{S} \times \mathcal{A} \to \{0,1\}^{(n+m) \times n}$ that represents the adjacency matrix of the sparsest local causal graph $\mathcal{G}^{\mathcal{L}}$ such that $\mathcal{L}$ is a neighborhood of $(s_t, a_t)$.[4] We apply $M$ to each transition to obtain local masks $\texttt{m}_1$ and $\texttt{m}_2$, compute their connected components, and swap independent components $\mathcal{G}_i$ and $\mathcal{G}_j$ (mutually disjoint and collectively exhaustive groups of connected components) between the transitions to produce a counterfactual proposal. We then apply $M$ to the counterfactual $(\tilde{s}_t, \tilde{a}_t)$ to validate the proposal—if the counterfactual mask $\tilde{\texttt{m}}$ shares the same graph partitions as $\texttt{m}_1$ and $\texttt{m}_2$, we accept the proposal as a CoDA sample. See Algorithm 1.

Note that masks $\texttt{m}_1$, $\texttt{m}_2$ and $\tilde{\texttt{m}}$ correspond to different neighborhoods $\mathcal{L}_1, \mathcal{L}_2$ and $\tilde{\mathcal{L}}$, so it is not clear that we are "within the bounds" of any model $\mathcal{M}^{\mathcal{L}}$ as was required in Subsection 2.2 for valid counterfactual reasoning. To correct this discrepancy we use the following proposition and additionally require the causal mechanisms (subgraphs) for independent components $\mathcal{G}_i$ and $\mathcal{G}_j$ to share structural equations in each local neighborhood: $f^{\mathcal{L}_1,i} = f^{\mathcal{L}_2,i} = f^{\tilde{\mathcal{L}},i}$ and $f^{\mathcal{L}_1,j} = f^{\mathcal{L}_2,j} = f^{\tilde{\mathcal{L}},j}$.[5] This makes our reasoning valid in the local subspace $\mathcal{L}^* = \mathcal{L}_1 \cup \mathcal{L}_2 \cup \tilde{\mathcal{L}}$. See Appendix A for proof.

**Proposition 1.** *The causal mechanisms represented by $\mathcal{G}_i, \mathcal{G}_j \subset \mathcal{G}$ are independent in $\mathcal{G}^{\mathcal{L}_1 \cup \mathcal{L}_2}$ if and only if $\mathcal{G}_i$ and $\mathcal{G}_j$ are independent in both $\mathcal{G}^{\mathcal{L}_1}$ and $\mathcal{G}^{\mathcal{L}_2}$, and $f^{\mathcal{L}_1,i} = f^{\mathcal{L}_2,i}, f^{\mathcal{L}_1,j} = f^{\mathcal{L}_2,j}$.*

Since CoDA only modifies data within local subspaces, this biases the resulting replay buffer to have more factorized transitions. In our experiments below, we specify the ratio of observed-to-counterfactual data heuristically to control this selection bias, but find that off-policy agents are reasonably robust to large proportions of CoDA-sampled trajectories. We leave a full characterization of the selection bias in CoDA to future studies, noting that knowledge of graph topology was shown to be useful in mitigating selection bias for causal effect estimation [5, 6].

**Inferring local factorization**   While the ground truth mask function $M$ may be available in rare cases as part of a simulator, the general case either requires a domain expert to specify an approximate causal model (as in goal relabeling and visual data augmentation) or requires the agent to learn the local factorization from data. Given how common independence due to physical separation of objects is, the former option will often be available. In the latter case, we note that the same data could also be used to learn a forward model. Thus, there is an implicit assumption in the latter case that learning the local factorization is easier than modeling the dynamics. We think this assumption is rather mild, as an accurate forward dynamics model would subsume the factorization, and we provide some empirical evidence of its validity in Section 4.

Learning the local factorization is similar to conditional causal structure discovery [81, 75, 67], conditioned on neighborhood $\mathcal{L}$ of $(s_t, a_t)$, except that the same structural equations must be applied globally (if the structural equations were conditioned on $\mathcal{L}$, Proposition 1 would fail). As there are many algorithms for general structure discovery [81, 75], and the arrow of time simplifies the inquiry

[27, 67], there may be many ways to approach this problem. For now, we consider a generalization of the global network mask approach used by MADE [25] (eq. 10) for autoregressive distribution modeling and GraN-DAG [44] (eq. 6) for causal discovery, which additionally conditions the mask on the current state and action.

This approach computes a locally conditioned network mask $M(s_t, a_t)$ by taking the matrix product of locally conditioned layer masks: $M(s_t, a_t) = \Pi_{\ell=1}^{L} M_\ell(s_t, a_t)$. This mask can be understood as an upper bound on the network's absolute Jacobian (see Appendix C). Again, there may be several models allow one to compute conditional layer masks. We tested two such models: a mixture of MLP experts and a single-head set transformer architecture [85, 47]. Each is described in more detail in Appendix C. Both are *trained* to model forward dynamics using an L2 prediction loss and induce a sparse network mask either via a sparsity penality (in case of the mixture of experts model) or via a sparse attention mechanism (in case of the set transformer). In preliminary experiments (Appendix C) we found that the set transformer performed better, and proceed to use it in our main experiments (Section 4). The set transformer uses the attention mask at each layer as the layer mask for that layer, so that the network mask is simply the product of the attention masks. Though trained to model forward dynamics, the CoDA models are used by the agent to *infer* local factorization rather than to directly sample future states as is typical in model-based RL. We found this produced reasonable results in the tested domains (below). See Appendix C for details. Future work should consider other approaches to inferring local structure such as graph neural networks [39, 13].

## 4 Experiments

Our experiments evaluate CoDA in the online, batch, and goal-conditioned settings, in each case finding that CoDA significantly improves agent performance as compared to non-CoDA baselines. Since CoDA only modifies an agent's training data, we expect these improvements to extend to other off-policy task settings in which the state space can be accurately disentangled. Below we outline our experimental design and results, deferring specific details and additional results to Appendix B.

**Standard online RL** We extend `Spriteworld` [89] to construct a "bouncing ball" environment (right), that consists of multiple objects (sprites) that move and collide within a confined 2D canvas. We use tasks of varying difficulty, where the agent must navigate $N \in \{1, 2, 3, 4\}$ of 4 sprites to their fixed target positions. The agent receives reward of $1/N$ for each of the $N$ sprites placed; e.g., the hardest task (Place 4) gives $1/4$ reward for each of 4 sprites placed. For each task, 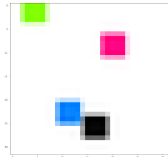 we use CoDA to expand the replay buffer of a TD3 agent [22] by about 8 times. We compare CoDA with a ground truth masking function (available via the `Spriteworld` environment) and learned masking function to the base TD3 agent, as well as a Dyna agent that generates additional training data by sampling from a model. For fair comparison, we use the same transformer used for CoDA masks for Dyna, which we pretrain using approximately 42,000 samples from a random policy. As in HER, we assume access to the ground truth reward function to relabel the rewards. The results in Figure 4 show that both variants of CoDA significantly improve sample complexity over the baseline. By contrast, the Dyna agent suffers from model bias, even though it uses the same model as CoDA.

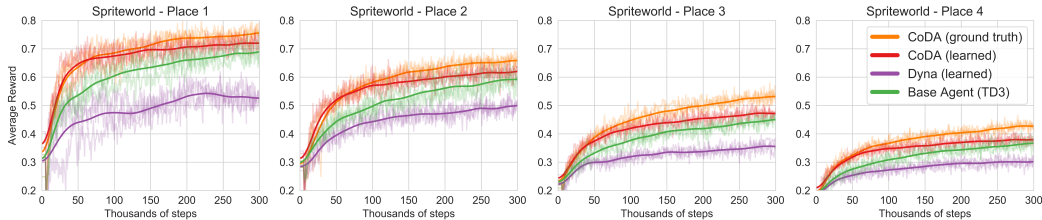

Figure 4: **Standard online RL** (3 seeds): CoDA with the ground truth mask always performs the best, validating our basic idea. CoDA with a pretrained model also offers a significant early boost in sample efficiency and maintains its lead over the base TD3 agent throughout training. Using the same model to generate data directly (a la Dyna [82]) performs poorly, suggesting significant model bias.

| $\mid\mathcal{D}\mid$ (1000s) | Real data 1ʀ | MBPO 1ʀ:1ᴍ | Ratio of Real:CoDA [:MBPO] data (ours) | | | |
|---|---|---|---|---|---|---|
| | | | 1ʀ:1ᴄ | 1ʀ:3ᴄ | 1ʀ:5ᴄ | 1ʀ:3ᴄ:1ᴍ |
| 25 | $13.2 \pm 0.7$ | $18.5 \pm 1.5$ | $43.8 \pm 2.8$ | $40.9 \pm 2.5$ | $38.4 \pm 4.9$ | $\mathbf{46.8} \pm 3.1$ |
| 50 | $22.8 \pm 3.0$ | $36.6 \pm 4.3$ | $66.6 \pm 3.8$ | $64.4 \pm 3.1$ | $62.5 \pm 3.5$ | $\mathbf{70.4} \pm 3.8$ |
| 75 | $43.2 \pm 4.9$ | $46.0 \pm 4.7$ | $73.4 \pm 2.8$ | $\mathbf{76.7} \pm 2.6$ | $75.0 \pm 3.4$ | $74.6 \pm 3.2$ |
| 100 | $63.0 \pm 3.1$ | $66.4 \pm 4.9$ | $77.8 \pm 2.0$ | $\mathbf{82.7} \pm 1.5$ | $76.6 \pm 3.0$ | $73.7 \pm 2.9$ |
| 150 | $77.4 \pm 1.2$ | $72.6 \pm 5.6$ | $82.2 \pm 1.8$ | $\mathbf{85.8} \pm 1.4$ | $84.2 \pm 1.0$ | $79.7 \pm 3.6$ |
| 250 | $78.2 \pm 2.7$ | $77.9 \pm 2.4$ | $85.0 \pm 2.9$ | $\mathbf{87.8} \pm 1.8$ | $87.0 \pm 1.0$ | $78.3 \pm 4.9$ |

Table 1: **Batch RL** (10 seeds): Mean success ($\pm$ standard error, estimated using 1000 bootstrap resamples) on `Pong` environment. CoDA with learned masking function more than doubles the effective data size, resulting in a 3x performance boost at smaller data sizes. Note that a 1ʀ:5ᴄ Real:CoDA ratio performs slightly worse than a 1ʀ:3ᴄ ratio due to distributional shift (Remark 3.3).

**Batch RL**    A natural setting for CoDA is batch-constrained RL, where an agent has access to an existing transition-level dataset, but cannot collect more data via exploration [21, 48]. This makes any additional, high quality data invaluable. Another reason why CoDA is attractive in this setting is that there is no *a priori* reason to prefer the given batch data distribution to a counterfactual one. For this experiment we use a continuous control `Pong` environment based

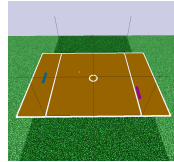

on `RoboschoolPong` [40]. The agent must hit the ball past the opponent, receiving reward of +1 when the ball is behind the opponent's paddle, -1 when the ball is behind the agent's paddle, and 0 otherwise. Since our transformer model performed poorly when used as a dynamics model, our Dyna baseline for batch RL adopts a state-of-the-art architecture [34] that employs a 7-model ensemble (MBPO). We collect datasets of up to 250,000 samples from an pretrained policy with added noise. For each dataset, we train both mask and reward functions (and in case of MBPO, the dynamics model) on the provided data and use them to generate different amounts of counterfactual data. We also consider combining CoDA with MBPO, by first expanding the dataset with MBPO and then applying CoDA to the result. We train the same TD3 agent on the expanded datasets in batch mode for 500,000 optimization steps. The results in Table 1 show that with only 3 state factors (two paddles and ball), applying CoDA is approximately equivalent to doubling the amount of real data.

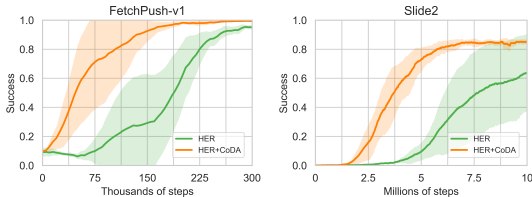

Fig. 5: **Goal-conditioned RL** (5 seeds): In `FetchPush` and the challenging `Slide2` environment, a HER agent whose dataset has been enlarged with CoDA approximately doubles the sample efficiency of the base HER agent.

**Goal-conditioned RL**    As HER [1] is an instance of prioritized CoDA that greatly improves sample efficiency in sparse-reward tasks, can our unprioritized CoDA algorithm further improve HER agents? We use HER to relabel goals on real data only, relying on random CoDA-style goal relabeling for CoDA data. After finding that CoDA obtains state-of-the-art results in `FetchPush-v1` [71], we show that CoDA also accelerates learning in a novel and significantly more

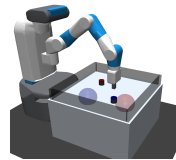

challenging `Slide2` environment, where the agent must slide two pucks onto their targets (Figure 5). For this experiment, we specified a heuristic mask using domain knowledge ("objects are disentangled if more than 10cm apart") that worked in both `FetchPush` and `Slide2` despite different dynamics.

## 5   Related Work

Factored MDPs [28, 29, 90] consider MDPs where state variables are only influenced by a fixed subset of "parent" variables at the previous timestep. The notion of "context specific independence" (CSI), which was used to compactly represent single factors of a Bayes net [10] or MDP [9] for efficient inference and model storage,[6] is closely related to the local factorizations we study in this paper. CSI can be understood as going one step beyond CoDA, exploiting not only knowledge of

the local factorization, but also the structural equations at play in the local factorization; CSI could be leveraged for model-based RL approaches where faithful models of factored dynamics can be realized. Object-oriented and relational approaches to RL and prediction [15, 26, 39, 50, 93, 94] represent the dynamics as a set of interacting entities. Factored actions and policies have been used to formulate dimension-wise policy gradient baselines in standard and multi-agent settings [19, 54, 91].

A growing body of work applies causal reasoning the RL setting to improve sample efficiency, interpretability and learn better representations [55, 57, 74]. Particularly relevant is the work by Buesing et al. [12], which improves sample efficiency by using a causal model to sample counterfactual trajectories, thereby reducing variance of off-policy gradient estimates in a guided policy search framework. These counterfactuals use coarse-grained representations at the trajectory level, while our approach uses factored representations within a single transition. Batch RL [48, 21, 58] and more generally off-policy RL [88, 60] are counterfactual by nature, and are particularly important when it is costly or dangerous to obtain on-policy data [84]. The use of counterfactual goals to accelerate learning goal-conditioned RL [36, 76, 71] is what inspired our local CoDA algorithm.

Data augmentation is also widely used in supervised learning, and is considered a required best practice in high dimensional problems [42, 46, 66]. Heuristics for data augmentation often encode a causal invariance statement with respect to certain perturbations on the inputs. Thus model performance on counterfactual/augmented data can be seen as a measure of *robustness*. Assuring that models perform robustly in this sense is relevant to applications where *fairness* is a concern, as counterfactuals can be used to achieve robust performance and debias data [23, 64, 8].

## 6 Conclusion

In this paper we proposed a local causal model (LCM) framework that captures the benefits of decomposition in settings where the global causal model is densely connected. We used our framework to design a local Counterfactual Data Augmentation (CoDA) algorithm that expands available training data with counterfactual samples by stitching together locally independent subsamples from the environment. Empirically, we showed that CoDA can more than double the sample efficiency and final performance of reinforcement learning agents in locally factored environments.

There are several interesting avenues for future work. First, the sizable gap between ground truth and learned CoDA in our Spriteworld results suggest there is room for improvement in our approach to learning the masking function. Second, we have applied CoDA in a random, unprioritized fashion, but past work [1, 77] suggests there is significant benefit to prioritization. Third, we have applied CoDA in a way that might be considered model-free, insofar as we reuse subsamples from the environment dynamics rather than generating samples using our models of the causal mechanims. However, our LCM formalism allows for mixing model-free and model-based reasoning, which could further improve sample efficiency. Fourth, we used fully-observable, disentangled state spaces with a fixed top-level decomposition, but ultimately we would like to deploy CoDA in partially observable, entangled settings (e.g. RL from pixels) with multiple dynamic, multi-level decompositions [32, 17, 92]. Unsupervised learning of factorized latent representations is an active area of research [16, 39, 52, 53, 89], and it would be interesting to combine these methods with CoDA. Finally, it would be interesting to explore applications of our LCM framework to other areas such as interpretability [56, 59], exploration [61, 86], and off-policy evaluation [84].

## Acknowledgments and Disclosure of Funding

We thank Jimmy Ba, Harris Chan, Seyed Kamyar Seyed Ghasemipour, James Lucas, David Madras, Yuhuai Wu and Lunjun Zhang for helpful comments and discussions. We also thank the anonymous reviewers for their feedback, which improved the final manuscript. SP is supported by an NSERC PGS-D award. EC is a student researcher at Google Brain in Toronto. Resources used in preparing this research were provided, in part, by the Province of Ontario, the Government of Canada through CIFAR, and companies sponsoring the Vector Institute (`https://vectorinstitute.ai/partners`).

## Broader Impact

**Considerations related to counterfactual reasoning**   CoDA transforms the observational data distribution into a counterfactual one. This incurs several risks and benefits, listed below.

- Modern machine learning models and reinforcement learning agents often generalize poorly under distributional shift (sometimes called "covariate shift") [49, 14]. CoDA has the potential to produce out-of-distribution data, that would never show up in the observational distribution. Thus, care should be taken when applying agent modules that have been trained only on observational data to counterfactual data, as their performance could decline sharply, thereby creating safety risks. We anticipate that work on uncertainty will be essential to controlling the risks associated with distributional shift [80].

- The fact that CoDA creates distributional shift can also provide benefits in the form of distributional robustness and fairness. Training on out-of-distribution data can make models more robust [79], increasing their trustworthiness and practical applicability. As noted in Section 5, counterfactual reasoning can be leveraged in areas where *fairness* is a concern. For example, [64] propose to reduce gender bias in natural language processing by generating counterfactual sentences with swapped gender pronouns. We anticipate that a version of CoDA that prioritizes fairness concerns in a similar manner could be applied in the reinforcement learning and computer vision contexts.

- In reinforcement learning specifically, a shift in data distribution requires either (1) the use of an off-policy algorithm, or (2) high variance off-policy corrections for on-policy algorithms. In the former case, it should be noted that in case of function approximation, even off-policy algorithms may be negatively affected by large shifts in their training distribution [83, 20]. More work is needed to quantify these effects and their implications for agent performance.

**Improving RL in batch-constrained settings**   In many settings, such as medicine and education, obtaining large quantities of observational data using a random policy is prohibitively expensive and/or unethical [84, 60]. As such, agents that can efficiently learn effective policies from batch-constrained data are needed, as are accurate ways to estimate agent performance from off-policy data [58, 62, 73]. We see CoDA as complementary to both goals, as subspace swapping is a powerful tool to produce large quantities of counterfactual data given a modest observational dataset. However, subspace swapping alone may be insufficient to generate plausible "exploratory" data for evaluating and learning new policies. For example, medical records from certain demographic groups may be unavailable or improperly collected/labeled. It is conceivable that a CoDA with a suitable prioritization scheme could compensate for such sample bias, but applied work in batch-constrained domains that characterizes the effect of sample bias on CoDA should nevertheless be carried out.

**General considerations related to artificial agency**   To the extent that CoDA is a general technique for improving the ability of artificial agents to achieve their goals, it inherits the potential risks and benefits associated with empowered artificial agency, including but not limited to:

 (a) the pursuit of misguided or dangerous goals, whether due to mispecification by a benevolent principal, the self-serving motives of its principals, or interference by malicious parties or other deviations from proper intents,

 (b) the unsafe and improper pursuit of goals due to poor modeling or representation, resource constraints and lack of capacity, constraint mispecification, partial observability, or inadequate encoding and understanding of human values, and

 (c) improvements to capital processes and automation of human labor, which could improve economic efficiency and raise the overall social welfare, but also run the risk of increased inequality, workforce displacement, and technological unemployment.

The risk associated with points (a) and (b) may be exacerbated in case of CoDA due to the risks associated with counterfactual reasoning outlined above: to the extent that CoDA is done with a poorly fit local factorization model, or with a local factorization model that does not generalize well to the counterfactual distribution, this could cause the agent to pursue poorly formulated counterfactual (imagined) goals, or create causally invalid data that hurts agent performance.

## Footnotes

[1]Code available at `https://github.com/spitis/mrl`

[2]Thus parentage does describe knock-on effects, e.g. $V_1$ on $V_3$ in the Markov chain $V_1 \rightarrow V_2 \rightarrow V_3$.

[3]As a trivial example, if $f^i$ is a function of binary variable $V^j$, and $\mathcal{L} = \{(s, a) \,|\, V^j = 0\}$, then $f^{\mathcal{L},i}$ is not a function of $V^j$ (which is now a constant), and there is no longer an edge from $V^j$ to $V^i$ in $\mathcal{G}^{\mathcal{L}}$.

[4]If Jacobian $\partial P / \partial x$ exists at $x = (s_t, a_t)$, the ground truth $M(s_t, a_t)$ equals $|(\partial P / \partial x)^T| > 0$.

[5]To see why this is not trivially true, imagine there are two rooms, one of which is icy. In either room the ground conditions are locally independent of movement dynamics, but not so if we consider their union.

[6]Note that these methods were proposed to efficiently encode conditional probability tables at the graph nodes, requiring that all variables considered be discrete; CoDA on the other hand works in continuous tasks.

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
