[Supplementary Material 1]

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

_{t[+1]}^i\}_{i=0}^{2n+m} = \{S_t^1 \ldots S_t^n, A_t^1 \ldots A_t^m, S_{t+1}^1 \ldots S_{t+1}^n\}$ are the nodes (variables) of $\mathcal{G}$.

- $U_t = \{U_{t[+1]}^i\}_{i=0}^{2n+m}$ is a set of noise variables, one for each $V^i$, determined by the initial state, past actions, and environment stochasticity. We assume that noise variables at time $t+1$ are independent from other noise variables: $U_{t+1}^i \perp\!\!\!\perp U_{t[+1]}^j \forall i, j$. The instance $u = (u^1, u^2, \ldots, u^{2n+m})$ of $U_t$ denotes an individual realization of the noise variables.

- $\mathcal{F} = \{f^i\}_{i=0}^{2n+m}$ is a set of functions ("structural equations") that map from $U_{t[+1]}^i \times \mathrm{Pa}(V_{t[+1]}^i)$ to $V_{t[+1]}^i$, where $\mathrm{Pa}(V_{t[+1]}^i) \subset V_t \setminus V_{t[+1]}^i$ are the parents of $V_{t[+1]}^i$ in $\mathcal{G}$; hence each $f^i$ is associated with the set of incoming edges to node $V_{t[+1]}^i$ in $\mathcal{G}$; see, e.g., Figure 2 (center).

Note that while $V_t$, $U_t$, and $\mathcal{M}_t$ are indexed by $t$ (their distributions change over time), the structural equations $f^i \in \mathcal{F}$ and causal graph $\mathcal{G}$ represent the global transition function $P$ and apply at all times $t$. To reduce clutter, we drop the subscript $t$ on $V$, $U$, and $\mathcal{M}$ when no confusion can arise.

Critically, we require the set of edges in $\mathcal{G}$ (and thus the number of inputs to each $f^i$) to be *structurally minimal* ([67], Remark 6.6).

**Assumption** (Structural Minimality). *$V^j \in Pa(V^i)$ if and only if there exists some $\{u^i, v^{-ij}\}$ with $u^i \in range(U^i), v^{-ij} \in range(V \setminus \{V^i, V^j\})$ and pair $(v_1^j, v_2^j)$ with $v_1^j, v_2^j \in range(V^j)$ such that $v_1^i = f^i(\{u^i, v^{-ij}, v_1^j\}) \neq f^i(\{u^i, v^{-ij}, v_2^j\}) = v_2^i$.*

Intuitively, structural minimality says that $V^j$ is a parent of $V^i$ if and only if setting the value of $V^j$ can have a nonzero *direct* effect[2] on the child $V^i$ through the structural equation $f^i$. The structurally minimal representation is unique [67].

Given structural minimality, we can think of edges in $\mathcal{G}$ as representing global causal dependence. The probability distribution of $S_{t+1}^i$ is fully specified by its parents $\mathrm{Pa}(S_{t+1}^i)$ together with its noise variable $U_i$; that is, we have $P(S_{t+1}^i \mid S_t, A_t) = P(S_{t+1}^i \mid \mathrm{Pa}(S_{t+1}^i))$ so that $S_{t+1}^i \

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

# A  Proof of Proposition 1

**Lemma 1.** *If $V^j \in Pa^{\mathcal{L}}(V^i)$ in DAG $\mathcal{G}^{\mathcal{L}}$ of (local) causal model $\mathcal{M}^{\mathcal{L}}$, and $\mathcal{L} \subset \mathcal{X}$, then $V^j \in Pa^{\mathcal{X}}(V^i)$ in DAG $\mathcal{G}^{\mathcal{X}}$ corresponding to causal model $\mathcal{M}^{\mathcal{X}}$.*

*Proof.* By minimality, there exist $\{u^i, v^{-j}, v_1^j\}$ and $\{u^i, v^{-j}, v_2^j\}$ with $v^{-j} \in Pa^{\mathcal{L}}(V^i) \backslash V^j$ for which $f^i(\{u^i, v^{-j}, v_1^j\}) \neq f^i(\{u^i, v^{-j}, v_2^j\})$. Expand $\{v^{-j}, v_1^j\}$ and $\{v^{-j}, v_2^j\}$ to $(s_1, a_1), (s_2, a_2) \in \mathcal{L}$ (with any values of other components). But $\mathcal{L} \subset \mathcal{X}$, so $(s_1, a_1), (s_2, a_2) \in \mathcal{X}$ and it follows from minimality in $\mathcal{X}$ that $V^j \in Pa^{\mathcal{X}}(V^i)$. $\qquad\square$

**Corollary 1.** *If $\mathcal{L} \subset \mathcal{X}$, $\mathcal{G}^{\mathcal{L}}$ is sparser (has fewer edges) than $\mathcal{G}^{\mathcal{X}}$.*

**Proposition 1.** *The causal mechanisms represented by $\mathcal{G}_i, \mathcal{G}_j \subset \mathcal{G}$ are independent in $\mathcal{G}^{\mathcal{L}_1 \cup \mathcal{L}_2}$ if and only if $\mathcal{G}_i$ and $\mathcal{G}_j$ are independent in both $\mathcal{G}^{\mathcal{L}_1}$ and $\mathcal{G}^{\mathcal{L}_2}$, and $f^{\mathcal{L}_1, i} = f^{\mathcal{L}_2, i}, f^{\mathcal{L}_1, j} = f^{\mathcal{L}_2, j}$.*

*Proof.* ($\Rightarrow$) If $\mathcal{G}_i$ and $\mathcal{G}_j$ are independent in $\mathcal{G}^{\mathcal{L}_1 \cup \mathcal{L}_2}$, independence in $\mathcal{G}^{\mathcal{L}_1}$ and $\mathcal{G}^{\mathcal{L}_2}$ follows from Corollary 1. That $f^{\mathcal{L}_1, i} = f^{\mathcal{L}_2, i}$ (and $f^{\mathcal{L}_1, j} = f^{\mathcal{L}_2, j}$), on their shared domain, follows since each is a restriction of the same function $f^{\mathcal{L}_1 \cup \mathcal{L}_2, i}$ (or $f^{\mathcal{L}_1 \cup \mathcal{L}_2, j}$).

($\Leftarrow$) Suppose $\mathcal{G}_i$ and $\mathcal{G}_j$ are independent in $\mathcal{G}^{\mathcal{L}_1}$ and $\mathcal{G}^{\mathcal{L}_2}$ but not $\mathcal{G}^{\mathcal{L}_1 \cup \mathcal{L}_2}$. By the definition of independence applied to $\mathcal{G}^{\mathcal{L}_1 \cup \mathcal{L}_2}$, we have that, without loss of generality, there is a $V_i \in \mathcal{G}_i, V_j \in \mathcal{G}_j$ with $V_j \in Pa^{\mathcal{L}_1 \cup \mathcal{L}_2}(V_i)$. Then, from the definition of minimality, it follows that there exist $(s_1, a_1), (s_2, a_2) \in \mathcal{L}_1 \cup \mathcal{L}_2$ that differ only in the value of $V_j$, and $u_i \in \text{range}(U_i)$ for which $f^i(s_1, a_1, u_i) \neq f^i(s_2, a_2, u_i)$.

Clearly, if $(s_1, a_1)$ and $(s_2, a_2)$ are both in $\mathcal{L}_1$ (or $\mathcal{L}_2$), there will be an edge from $V_j$ to $V_i$ in $\mathcal{G}^{\mathcal{L}_1}$ (or $\mathcal{G}^{\mathcal{L}_2}$) and the claim follows by contradiction. Thus, the only interesting case is when, without loss of generality, $(s_1, a_1) \in \mathcal{L}_1$ and $(s_2, a_2) \in \mathcal{L}_2$. The key observation is that $(s_1, a_1)$ and $(s_2, a_2)$ differ only in the value of node $V_j \notin \mathcal{G}_i$: since $\mathcal{G}_i$ is an independent causal mechanism in both $\mathcal{G}^{\mathcal{L}_1}$ and $\mathcal{G}^{\mathcal{L}_2}$ and the parents of $V_i$ take on the same values in each, we have that $f^i(s_1, a_1, u_i) = f^{i, \mathcal{L}_1}(s_1, a_1, u_i) = f^{i, \mathcal{L}_2}(s_2, a_2, u_i) = f^i(s_2, a_2, u_i)$ and the claim follows by contradiction. $\quad\square$

# B  Additional Experiment Details

This section provides training details for the experiments discussed in Section 4 as well as some additional results. Code is available at `https://github.com/spitis/mrl` [69].

## B.1  Online RL

Here we detail the procedure used in the Online RL experiments in the Spriteworld environment.

**Implementation**  We work with the original TD3 codebase, architecture, and hyperparameters (except batch size; see below), and focus our efforts solely on modifying the agent's training distribution.

**Environment**  We extend the base Spriteworld framework [89] as follows:

1. We add basic collisions, wherein a sprites velocity is reversed if it would overlap with another sprite on the next frame. This induces a local factorization and prevents a global factorization from being applicable.

2. We use a new continuous action space that consists of a 2-dimensional "click" on the canvas. The click changes the velocity of the closest sprite according to distance and angle between the click and the sprite's center.

3. We add a disentangled state renderer that returns the position and velocity of each sprite (a total of 16-dimensions in tasks with four sprites).

4. We add a mask renderer that returns the ground truth masking function. This allows us to evaluate our masking function in Appendix C.

5. We add a suite of partial and sparse reward tasks that we use for experiments. The tasks vary in difficulty along the following two dimensions

- **Number of shapes:** The agent must place $N \in \{1, 2, 3, 4\}$ of the 4 sprites at their respective target (one of four quadrants).
- **Sparsity of reward signal:** The agent receives reward either *Partial*, $\frac{1}{N}$ reward for each of the $N$ sprites placed near its target (up to a maximum reward of 1); or *Sparse*, reward of 1 is only observed when *all* $N$ sprites reach their target.

For example, hardest task—"sparse reward, place 4" in the results—requires placing 4 shapes with sparse reward. Episodes run for 50 timesteps, and there is no termination signal (i.e., the environment is continuous from the agent's perspective). We also tried a variant with dense rewards, but found that the baseline agent was able to easily solve this task and no benefit was conferred by data augmentation.

These extensions will included with the release of our code.

**Data augmentation**   Every 1000 environment steps, we sample 2000 pairs of random transitions from the agent's replay buffer, and apply CoDA to produce a maximum of 5 unique CoDA samples per pair. We apply two forms of CoDA, using (1) an oracle / ground truth mask function that we back out of the simulator, and (2) the mask of a pre-trained transformer model (see Section C). The mask function was trained using approximately 42,000 samples from a random policy (5/6 of 50,000, with the rest of the data used for validation). CoDA samples are added to a second CoDA replay buffer. For purposes of this experiment both buffers are have effectively infinite capacity (they are never filled). During training, the agent's batches are sampled proportionally from the real and CoDA replay buffers (this means that approximately 7/8 of the data that the agent trains on is counterfactual).

**Baselines**   In addition to the base TD3 agent and CoDA, we also use the transformer model that is used as a mask function to generate data by performing forward rollouts with a random policy, as in Dyna [82]. So that this baseline produces approximately the same number of samples as CoDA with the learned mask, we roll the model out for 5 steps from 1500 random samples from the replay buffer, again every 1000 environment steps. This produces 7500 model-based samples for every 1000 environment transitions.

As an additional model-based Dyna baseline, we also fit a plain feedforward neural network with 2 hidden layers of 256 ReLU units as a feedforward model. We found that this forward model performed much worse than our transformer-based dyna baseline. The results are shown in Figure 6.

Figure 6: **Average reward in Partial reward setting with MLP-based Dyna baseline.**

Use of the transformer mask function for CoDA requires setting the threshold value $\tau$, which we do by monitoring accuracy and F1 scores for sparsity prediction (as discussed in Appendix C) on validation data, ultimately using the value $\tau = 0.05$.

**Batch Size**   Since CoDA samples are plentiful we increase the agent's batch size from 256 to 1000 to allow it to train on more environment samples in the same number of training steps. We found that this slightly improved the performance of the base TD3 agent. An increase in batch size also allows the agent to see more of its own on-policy data in the face of many off-policy CoDA samples.

**Additional results in sparse reward task variants**   In addition to the partial reward tasks described in Section 4, we also tested CoDA in four sparse reward tasks of varying difficulty. These are the same as the partial reward tasks, except that a sparse reward of 1 is granted only when *all* N sprites

Figure 7: **Average reward on Sparse reward bouncing ball environment.** As in the partial reward case (Figure 4), we observe that CoDA agents outperform the other agents (except in Place 4, where no agent achieves any reward).

are in their target locations. While these tasks were much harder (and perhaps impossible in the case of Place 4 due to moving sprites), as shown in Figure 7, the CoDA agents maintain a clear advantage.

**Results plot** The results plot shows the 3 seeds in reduced opacity together with their smoothed mean.

## B.2 Batch RL

Here we detail the procedure used in the Batch RL experiments in the `Pong` environment.

**Implementation** This experiment works with a different codebase than the Spriteworld experiment, in order to simplify the use of CoDA in a Batch RL setting. Our experiment first builds the agent's dataset (consisting of real data, dyna data and/or CoDA data), then instantiates a TD3 agent by filling its replay buffer with the dataset. The replay buffer is always expanded to include the entire enlarged dataset (for the 5x CoDA ratio at 250,000 data size this means the buffer has 1.5E6 experiences). The agent is run for 500,000 optimization steps.

**Hyperparameters** We used similar hyperparameters to the original TD3 codebase, with the following differences:

- We use a discount factor of $\gamma = 0.98$ instead of 0.99.
- Since Pong is a sparse reward task with $\gamma = 0.98$, we clip critic targets to $(-50, 50)$.
- We use networks of size $(128, 128)$ instead of $(256, 256)$.
- As for our Spriteworld experiment, we use a batch size of 1000.

**Environment** We base our `Pong` environment on `RoboSchoolPong-v1` [40]. The original environment allowed the ball to teleport back to the center after one of the players scored, offered a small dense reward signal for hitting the ball, and included a stray "timeout" feature in the agent's state representation. We fix the environment so that the ball does not teleport, and instead have the episode reset every 150 steps, and also 10 steps after either player scores. The environment is treated as continuous and never returns a done signal that is not also accompanied by a `TimeLimit.truncated` indicator [11]. We change the reward to be strictly sparse, with reward of $\pm1$ given when the ball is behind one of the players' paddles. Finally, we drop the stray "timeout" feature, so that the state space is 12-dimensional, where each set of 4 dimensions is the x-position, y-position, x-velocity, and y-velocity of the corresponding object (2 paddles and one ball).

The opponent in the environment uses a small pretrained policy that was included in the original environment [40] (the effectiveness of this policy was unaffected by the removal of the timeout feature). To collect the batch dataset, we use the same policy as an "expert" policy, but replace 50% of its actions by random actions.

**Training the CoDA model** Without access to a ground truth mask, we needed to train a masking function C to identify local disentanglement. We also forewent the ground truth reward, instead training our own reward classifier. In each case we used the batch dataset given, and so we trained different models for each random seed. For our masking model, we stacked two single-head

transformer blocks (without positional encodings) and used the product of their attention masks as the mask. Each block consists of query $Q$, key $K$, and value $V$ networks that each have 3-layers of 256 neurons each, with the attention computed as usual [85, 47]. The transformer is trained to minimize the L2 error of the next state prediction given the current state and action as inputs. The input is disentangled, and so has shape (`batch_size, num_components, num_features`). In each row (component representation) of each sample, features corresponding to other components are set to zero. The transformer is trained for 2000 steps with a batch size of 256, Adam optimizer [38] with learning rate of 3e-4 and weight decay of 1e-5. For our reward function we use a fully-connected neural network with 1 hidden layer of 128 units. The reward network accepts an $(s, a, s')$ tuple as input (not disentangled) and outputs a softmax over the possible reward values of $[-1, 0, 1]$. It is trained for 2000 steps with a batch size of 512, Adam optimizer [38] with learning reate of 1e-3 and weight decay of 1e-4. All hyperparameters were rather arbitrary (we used the default setting, or in case it did not work, the first setting that gave reasonable results, as was determined by inspection). To ensure that our model and reward functions are trained appropriately (i.e., do not diverge) for each seed, we confirm that the average loss of the CoDA model is below 0.005 at the training and that the average loss of the reward model is below 0.1, which values were found by inspection of a prototype run. These conditions were met by all seeds.

When used to produce masks, we chose a threshold of $\tau = 0.02$ by inspection, which seemed to produce reasonable results. A more principled approach would do cross-validation on the available data, as we did for Spriteworld (Appendix C).

**Tested configurations** Table 1 reports a subset of our tested configurations. We report our results in full below. We considered the following configurations:

1. **Real data only**.
2. **CoDA + real data**: after training the CoDA model, we expand the base dataset by either 2, 3, 4 or 6 times.
3. **Dyna (using CoDA model)**: after training the CoDA model, we use it as a forward dynamics model instead of for CoDA; we use 1-step rollouts with random actions from random states in the given dataset to expand the dataset by 2x. We also tried with 5-step rollouts, but found that this further hurt performance (not shown). Note that Dyna results use only 5 seeds.
4. **Dyna (using MBPO model)**: as the CoDA model exhibits significant model bias when used as a forward dynamics model, we replicate the state-of-the-art model-based architecture used by MBPO [34] and use it as a forward dynamics model for Dyna; we experimented with 1-step and 5-step rollouts with random actions from random states in the given dataset to expand the dataset by 2x. This time we found that the 5-step rollouts do better, which we attribute to the lower model bias together with the ability to create a more diverse dataset (1-step not shown). The MBPO model is described below. We use the same reward model as CoDA to relabel rewards for the MBPO model, which only predicts next state.
5. **MBPO + CoDA**: as MBPO improved performance over the baseline (real data only) at lower dataset sizes, we considered using MBPO together with CoDA. We use the base dataset to train the MBPO, CoDA, and reward models, as described above. We then use the MBPO model to expand the base dataset by 2x, as described above. We then use the CoDA model to expand *the expanded dataset* by 3x the original dataset size. Thus the final dataset is 5x as large as the original dataset (1 real : 1 MBPO : 3 CoDA).

All configurations alter only the training dataset, and the same agent architecture/hyperparameters (reported above) are used in each case.

**MBPO model** Since using the CoDA model for Dyna harms rather than helps, we consider using a stronger, state-of-the-art model-based approach. In particular, we adopt the model used by Model-Based Policy Optimization [34]. This model consists of a size 7 ensemble of neural networks, each with 4 layers of 200 neurons. We use ReLU activations, Adam optimizer [38] with weight decay of 5e-5, and have each network output a the mean and (log) diagonal covariance of a multi-variate Gaussian. We train the networks with a maximum likelihood loss. To sample from the model, we choose an ensemble member uniformly at random and sample from its output distribution, as in [34].

**Full results** See Table 2 for results.

| $\|\mathcal{D}\|$ (1000s) | Real data 1ʀ | Dyna 1ʀ:1ᴍ | MBPO 1ʀ:1ᴍ | Ratio of Real:CoDA [:MBPO] data (ours) | | | | |
|---|---|---|---|---|---|---|---|---|
| | | | | 1ʀ:1ᴄ | 1ʀ:2ᴄ | 1ʀ:3ᴄ | 1ʀ:5ᴄ | 1ʀ:3ᴄ:1ᴍ |
| 25 | $13.2 \pm 0.7$ | $12.2 \pm 1.4$ | $18.5 \pm 1.5$ | $43.8 \pm 2.7$ | $41.6 \pm 3.7$ | $40.9 \pm 2.5$ | $38.4 \pm 4.9$ | $\mathbf{46.8} \pm 3.2$ |
| 50 | $22.8 \pm 3.2$ | $17.4 \pm 3.0$ | $36.6 \pm 4.0$ | $66.6 \pm 3.7$ | $58.4 \pm 4.7$ | $64.4 \pm 3.1$ | $62.5 \pm 3.5$ | $\mathbf{70.4} \pm 3.7$ |
| 75 | $43.2 \pm 4.7$ | $23.3 \pm 4.7$ | $46.0 \pm 4.5$ | $73.4 \pm 2.8$ | $71.6 \pm 3.8$ | $\mathbf{76.7} \pm 2.6$ | $75.0 \pm 3.3$ | $74.6 \pm 3.3$ |
| 100 | $63.0 \pm 3.0$ | $25.9 \pm 7.4$ | $66.4 \pm 5.2$ | $77.8 \pm 2.0$ | $77.4 \pm 1.8$ | $\mathbf{82.7} \pm 1.5$ | $76.6 \pm 3.0$ | $73.7 \pm 2.9$ |
| 150 | $77.4 \pm 1.3$ | $34.1 \pm 1.5$ | $72.6 \pm 5.6$ | $82.2 \pm 1.7$ | $84.0 \pm 1.2$ | $\mathbf{85.8} \pm 1.4$ | $84.2 \pm 1.0$ | $79.7 \pm 3.5$ |
| 250 | $78.2 \pm 2.7$ | $44.2 \pm 4.0$ | $77.9 \pm 2.3$ | $85.0 \pm 2.8$ | $\mathbf{88.1} \pm 1.0$ | $87.8 \pm 1.7$ | $87.0 \pm 1.0$ | $78.3 \pm 5.0$ |

Table 2: Extended Batch RL results. Mean success ($\pm$ standard error, estimated using 1000 bootstrap resamples) on Pong environment. All results average over 10 seeds, except Dyna, which uses 5.

### B.3  Goal-conditioned RL

Here we detail the procedure used in the Goal-conditioned RL experiments on the Fetchpush-v1 and Slide2 environments.

**Implementation**  This experiment uses the same codebase as our Batch RL, which provides state-of-the-art baseline HER agents and will be released with the paper.

**Hyperparameters**  For Fetchpush-v1 we use the default hyperparameters from the codebase (see [70] for details on how they were selected), which outperform the original HER agents of [1, 71] and follow-up works. We do not tune the CoDA agent (but see additional CoDA hyperparameters below). They are as follows:

- Off-policy algorithm: DDPG [51]
- Hindsight relabeling strategy: futureactual_2_2 [68], using exclusively future [1] relabeling for the first 25,000 steps
- Optimizer: Adam [38] with default hyperparameters
- Batch size: 2000
- Optimization frequency: 1 optimization step every 2 environment steps after the 5000th environment step
- Target network updates: update every 10 optimization steps with a Polyak averaging coefficient of 0.05
- Discount factor: 0.98
- Action l2 regularization: 0.01
- Networks: 3x512 layer-normalized [4] hidden layers with ReLU activations
- Target clipping: (-50, 0)
- Action noise: 0.1 Gaussian noise
- Epsilon exploration [71]: 0.2, with an initial 100% exploration period of 10,000 steps
- Observation normalization: yes
- Buffer size: 1M

On Slide2 we tried to tune the baseline hyperparameters somewhat, but note that this is a fairly long experiment (10M timesteps) and so only a few settings were tested due to constraints. In particular, we considered the following modifications:

- Expanding the replay buffer to 2M (effective)
- Reducing the batch size to 1000 (effective)* (used for results)
- Using the future_4 strategy (agent fails to learn in 10M steps)
- Reducing optimization step frequency to 1 step every 4 environment steps (about the same performance)

We tried similar adjustments to our CoDA agent, but found the default hyperparameters (used for results) performed well. We found that the CoDA agent outperforms the base HER agent on all tested settings.

For CoDA, we used the following additional hyperparameters:

- CoDA buffer size: 3M

- Make CoDA data every: 250 environment steps

- Number of source pairs from replay buffer used to make CoDA data: 2000

- Number of CoDA samples per source pair: 2

- Maximum ratio of CoDA:Real data to train on: 3:1

**Environment**    On `FetchPush-v1` the standard state features include the relative position of the object and gripper, which entangles the two. While this could be dealt with by dynamic relabeling (as used for HER's reward), we simply drop the corresponding features from the state.

Figure 8: The `Slide2` environment.

`Slide2` has two pucks that slide on a table and bounce off of a solid railing. Observations are 40-dimensional (including the 6-dimensional goal), and actions are 4-dimensional. Initial positions and goal positions are sampled randomly on the table. During training, the agent gets a sparse reward of 0 (otherwise -1) if *both* pucks are within 5cm of their ordered target. At test time we count success as having both picks within 7.5cm of the target on the last step of the episode. Episodes last 75 steps and there is no done signal (this is intended as a continuous task).

**CoDA Heuristic**    For these experiments we use a hand-coded heuristic designed with domain knowledge. In particular, we assert that the action is always entangled with the gripper, and that gripper/action and objects (pucks or blocks) are disentangled whenever they are more than 10cm apart. This encodes independence due to physical separation, which we hypothesize is a very generally heuristic that humans implicitly rely on all the time. The pucks have a radius of 2.5cm and height of 4cm, and the blocks are 5cm x 5cm x 5cm, so this heuristic is quite generous / suboptimal. Despite being suboptimal, it demonstrates the ease with which domain knowledge can be injected via the CoDA mask: we need only a high precision (low false positive rate) heuristic—the recall is not as important. It is likely that an agent could learn a better mask that also takes into account velocity.

**Results plot**    The plot shows mean $\pm 1$ standard deviation of the smoothed data over 5 seeds.

## C    Inferring Local Factorization

Here we present several approaches to inferring the local factorization of subspaces, a crucial subroutine of CoDA. We note that in many cases where domain knowledge is available, simple heuristics may suffice, e.g. in our Goal-conditioned RL experiments discussed in Section 4 where a simple distance-based indicator function in the state space was used. However, as such heuristics may not be universally available, the question of whether data-driven approaches can successfully infer local factorization is of general interest. We note that the performance of CoDA will improve alongside future improvements in this inference task (motivating future work in this area), and that inferring the local factorization in general is an easier task than learning the environment dynamics.

We begin by presenting two methods for inferring local factorization, derived from variants of a next-state prediction task, which we here refer to as SANDy for Sparse Attention Neural Dynamics. To verify the merits of SANDy in the local factorization inference task, we evaluate two SANDy variants in controlled settings where the ground truth factorization is known: first in a synthetic Markov process (MDP without actions), and second in Spriteworld. This label information is used only to evaluate performance, and not to train the SANDy parameters. The SANDy-Transformer model was used for the Online RL experiments presented in Section 4.

## C.1 Methods

We propose two Sparse Attention for Neural Dynamics (SANDy) models. In each case we seek to learn a function (or mask) $M(s,a) \rightarrow \{0,1\}^{(|S|+|A|) \times |S|}$ whose output represents the adjacency matrix of the local causal graph, conditioned on the state and action. We note that $M(s,a)_{i,j} = 0$ alone is insufficient, in general, to determine the local subspace $\mathcal{L} \subset \mathcal{S} \times \mathcal{A}$, since there may be multiple disconnected subspaces $\mathcal{L}_k$ with $M_k(s,a)_{i,j} = 0$ whose union $\bigcup_k \mathcal{L}_k$ has $M_{\cup k}(s,a)_{i,j} = 1$. This can be resolved by our Proposition 1 if we also force the relevant structural equations to be the same. For now, we assume the mask determines the local subspace, and leave exploration of this possibility to future work. Empirically, we will see that our assumption is reasonable.

**SANDy-Mixture**  The first model is a mixture-of-MLPs model with an attention mechanism that is computed from the current state. Each component of the mixture is a neural dynamics model with sparse local dependencies. For a given component, the key idea is to train a neural dynamics model to predict the next state $h(s_t, a_t) \approx s_{t+1}$ and approximate the masking function by thresholding the (transpose of the) network Jacobian of $h$, $[\mathbf{J}(s,a)]_{i,j} = \frac{\delta}{\delta[s,a]_j}[h(s,a)]_i$. Intuitively, we can think of $\mathbf{J}$ as providing the first-order element-wise dependencies between the predicted next state and the network input. We then derive the local factorization by thresholding the absolute Jacobian

$$M_\tau(s,a) = \mathbb{1}(|\mathbf{J}(s,a)| > \tau), \tag{1}$$

where $\mathbb{1}(\cdot)$ represents the indicator function and $\tau$ is a threshold hyperparameter.

To estimate the network Jacobian, we note that for standard activation functions (sigmoid, tanh, relu), it can be bound from above by the matrix product of its weight matrices. To see this, let $h_\theta$ be an $L$-layer MLP parameterized by $\theta = (\mathbf{W}^{(1)}, \mathbf{b}^{(1)}, \cdots, \mathbf{W}^{(L)}, \mathbf{b}^{(L)})$ with activation $\sigma$ with bounded derivative $\sigma'(x) \leq 1$, and note that for each layer $\boldsymbol{h}^{(\ell)} = \sigma(\mathbf{W}^{(\ell)} \boldsymbol{h}^{(\ell-1)} + \mathbf{b}^{(\ell)})$ we have:

$$\left| \frac{d\boldsymbol{h}_j^{(\ell)}}{d\boldsymbol{h}_i^{(\ell-1)}} \right| \leq \left| \mathbf{W}_{ji}^{(\ell)} \right|.$$

Then, using the chain rule, triangle inequality, and the identity $|ab| = |a||b|$, we can compute:

$$\left| \frac{d\boldsymbol{h}_j^{(\ell)}}{d\boldsymbol{h}_i^{(\ell-2)}} \right| \leq \left| \frac{d}{d\boldsymbol{h}_i^{(\ell-2)}} \mathbf{W}_{j\cdot}^{(\ell)} \boldsymbol{h}^{(\ell-1)} \right| = \left| \mathbf{W}_{j\cdot}^{(\ell)} \cdot \frac{d\boldsymbol{h}^{(\ell-1)}}{d\boldsymbol{h}_i^{(\ell-2)}} \right|$$

$$\leq \sum_k \left| \mathbf{W}_{jk}^{(\ell)} \frac{d\boldsymbol{h}_k^{(\ell-1)}}{d\boldsymbol{h}_i^{(\ell-2)}} \right| \leq \left| \mathbf{W}_{j\cdot}^{(\ell)} \right| \cdot \left| \mathbf{W}_{\cdot i}^{(\ell-1)} \right|.$$

Expanding this out to multiple layers, we see that $\left| \mathbf{J}_\theta(s,a) \right| \leq \prod_{\ell \in [L]} |\mathbf{W}^{(\ell)}|$, as desired. We use this upper bound to approximate the network Jacobian of an MLP by setting $\hat{\mathbf{J}} = \prod_{\ell \in [L]} |\mathbf{W}^{(\ell)}|$. A similar idea is used by [25, 44], among others, to control element-wise input-output dependencies.

We use this static approximation $\hat{\mathbf{J}}$ to facilitate learning a sparse dynamic mask by specifying a mixture model $h(s,a) = \sum_i \alpha_\phi^{(i)}(s,a) h_\theta^{(i)}(s,a)$ over the environment dynamics (with $\sum_i \alpha_\phi^{(i)}(s,a) = 1 \; \forall \; (s,a)$), where each component $h_\theta^{(i)}$ is an MLP as specified above with a sparsity prior on its Jacobian bound $\hat{\mathbf{J}}^{(i)}$ to encourage sparse (i.e. well-factorized) local solutions. The dynamic mask is computed by first approximating the Jacobian, $\hat{\mathbf{J}}(s,a) = \sum_i \alpha_\phi^{(i)}(s,a) \hat{\mathbf{J}}^{(i)}$, then thresholding by $\tau$ as in Equation 1.

Note that we assume the mixture components $\alpha$ are a function of the current state and action alone; in other words the factorization (captured by the network Jacobian of each component) can be *locally inferred*. The next-state prediction is given by $\hat{\mathbf{s}}_{t+1} = (h_\theta^{(1)}(\mathbf{s}_t, \mathbf{a}_t), \cdots, h_\theta^{(N)}(\mathbf{s}_t, \mathbf{a}_t))^T \boldsymbol{\alpha}_\phi(\mathbf{s}_t, \mathbf{a}_t)$. To train the model, we optimize the objective:

$$\underset{\theta,\phi}{\text{minimize}} \; \frac{1}{|\mathcal{D}|} \left( \sum_{(\mathbf{s}_t, \mathbf{a}_t \mathbf{s}_{t+1}) \in \mathcal{D}} ||\mathbf{s}_{t+1} - h(\mathbf{s}_t, \mathbf{a}_t)||_2^2 \right) + \lambda_1 S(\theta) + \lambda_2 R(\phi) + \lambda_3 ||(\theta, \phi)||_2 \tag{2}$$

where $S(\theta) = \frac{1}{K} \sum_i |\hat{\mathbf{J}}_\theta^{(i)}|_1$ puts an $\ell_1$ prior on each mixture component to induce sparsity, and $R(\phi)$ encourages high entropy in the attention probabilities

$$R(\phi) = \frac{1}{|\mathcal{D}|} \sum_{\mathbf{s} \in \mathcal{D}} \sqrt{\frac{1}{N} \sum_{j \in [N]} [A_\phi(\mathbf{s}, \mathbf{a})]_j}.$$

We note that more sophisticated methods of computing Jacobians of neural networks–including architectural changes and optimization strategies [3]—have been proposed, and could in principle be used here as well.

**SANDy-Transformer**  As an alternative model, we use a transformer-like architecture that applies self-attention between a set of (potentially multi-dimensional) inputs [85, 47]. Our architecture is composed of a stacked self-attention blocks. Each block accepts a set of inputs $X = \{x_i \in \mathbb{R}^n\}$ and composes three functions of each input: query $Q : \mathbb{R}^n \to \mathbb{R}^d$, key $K : \mathbb{R}^n \to \mathbb{R}^d$, and value $V : \mathbb{R}^n \to \mathbb{R}^m$. The block returns a set of outputs $\{y_i \in \mathbb{R}^m\}$ of size $|X|$, each of which is computed as: $y_i = A_i^T V(X)$, where $A_i = \text{softmax}\left(\sum_j (Q(x_i)_j K(x_j)_i)\right)$ (note that $V(X)$ is a matrix of size $|X| \times m$). We approximate the block mask (approximate Jacobian) as $A = [A_1, A_2, \ldots, A_{|X|}] \in \mathbb{R}^{|X| \times |X|}$, and the full network mask as the product of the block masks (as in the SANDy-Mixture). We used two-layer MLPs for each function $K, Q, V$ in Spriteworld and three-layer MLPs in Pong.

To apply this architecture to our problem, we first embed each state and action component (single feature, or group of features) into $\mathbb{R}^n$ to produce a set of inputs $X$ and pass this through each stacked self-attention block to obtain a set of outputs $Y$. We then discard any output components that correspond to the action features to obtain the next state prediction and mask. The network $h$ is trained to minimize mean squared error:

$$\underset{\theta}{\text{minimize}} \frac{1}{|\mathcal{D}|} \sum_{(\mathbf{s}_t, \mathbf{a}_t \mathbf{s}_{t+1}) \in \mathcal{D}} ||\mathbf{s}_{t+1} - h(\mathbf{s}_t, \mathbf{a}_t)||_2^2. \tag{3}$$

Unlike the SANDy-Mixture, no sparsity regularizers are applied, as we found the sparsity induced by the softmax attention mechanism to be sufficient.

## C.2  Evaluation environments

**Synthetic Markov Processes**  We investigate the capacity of the SANDy-Mixture to learn simple factorized transition dynamics under a globally factored Markov Process (STATIONARYMP). Unlike the general MDP setting, no agent/policy is considered. However the ability to train an unconstrained dynamics model to approximate factorized environment dynamics is an important subtask within our overall approach. Assuming a spherical Gaussian prior over the initial states $\mathbf{s}^0 \in \mathbb{R}^9$, the STATIONARYMP is entirely specified by transition distribution $p(\mathbf{s}^{t+1}|\mathbf{s}^t)$. We assume that the state transitions factorize (globally) into three parts. Denoting by $\mathbf{s}_{n \cdots m}^t$ the $n$-th through $m$-th dimensions of the time $t$ state $\mathbf{s}^t$, we have:

$$p(\mathbf{s}_{1\ldots9}^{t+1}|\mathbf{s}_{1\ldots9}^t) = p(\mathbf{s}_{1\ldots4}^{t+1}|\mathbf{s}_{1\ldots4}^t)p(\mathbf{s}_{5\ldots7}^{t+1}|\mathbf{s}_{5\ldots7}^t)p(\mathbf{s}_{8,9}^{t+1}|\mathbf{s}_{8,9}^t).$$

In other words, we have a block-diagonal transition matrix comprising three blocks with sizes 4, 3, and 2, respectively. In our case, all transition factors are deterministic non-linear mappings, e.g. $p(\mathbf{s}_{1\ldots4}^{t+1}|\mathbf{s}_{1\ldots4}^t) = \delta(g_{1\ldots4}(\mathbf{s}_{1\ldots4}^t))$, with $g_{1\ldots4} : \mathbb{R}^4 \to \mathbb{R}^4$ is a randomly-initialized single-hidden-layer neural network with 32 hidden units and GELU nonlinearity [30]. In this case, we can alternatively express the deterministic dynamics via the transition function

$$\begin{aligned}
\mathbf{s}^{t+1} &= (\mathbf{s}_{1\ldots4}^{t+1}, \mathbf{s}_{5\ldots7}^{t+1}, \mathbf{s}_{8,9}^{t+1}) \\
&= (g_{1\ldots4}(\mathbf{s}_{1\ldots4}^t), g_{5\ldots7}(\mathbf{s}_{5\ldots7}^t), g_{8,9}(\mathbf{s}_{8,9}^t)). \qquad \text{(STATIONARYMP)}
\end{aligned}$$

We now turn to a more sophisticated MP with locally factored dynamics ($\epsilon$-NONSTATIONARYMP), and investigate whether the SANDy-Mixture can learn to recognize local disentanglement.

We refer to the component functions $g_{1\ldots4} : \mathbb{R}^4 \to \mathbb{R}^4$, $g_{5\ldots7} : \mathbb{R}^3 \to \mathbb{R}^3$, and $g_{8,9} : \mathbb{R}^2 \to \mathbb{R}^2$ used in the STATIONARYMP as *local* transitions. We now introduce *global* interactions via the global transition functions, $G_{1\ldots4} : \mathbb{R}^4 \to \mathbb{R}^9$, $G_{5\ldots7} : \mathbb{R}^4 \to \mathbb{R}^9$, and $G_{8,9} : \mathbb{R}^4 \to \mathbb{R}^9$, which respectively

map the local state factors onto the global state space[7]. This allows us to extend the STATIONARYMP by adding global state transitions to the local state transitions whenever the norm of the local state factors exceeds the value of a hyperparameter $\epsilon$ (lower $\epsilon$ indicates more global interaction). Denoting by $\mathbb{1}(\cdot)$ the indicator function, we have

$$
\begin{aligned}
\mathbf{s}^{t+1} &= \left[(\mathbf{s}_1^{t+1}, \mathbf{s}_2^{t+1}, \mathbf{s}_3^{t+1}, \mathbf{s}_4^{t+1}), (\mathbf{s}_5^{t+1}, \mathbf{s}_6^{t+1}, \mathbf{s}_7^{t+1}), (\mathbf{s}_8^{t+1}, \mathbf{s}_9^{t+1})\right] \\
&= \left[g_{1\ldots4}(\mathbf{s}_{1\ldots4}^t), g_{5\ldots7}(\mathbf{s}_{5\ldots7}^t), g_{8,9}(\mathbf{s}_{8,9}^t)\right] \\
&\quad + G_{1\ldots4}(\mathbf{s}_{1\ldots4}^t)\mathbb{1}(||\mathbf{s}_{1\ldots4}^t||_2 > \epsilon) \\
&\quad + G_{5\ldots7}(\mathbf{s}_{5\ldots7}^t)\mathbb{1}(||\mathbf{s}_{5\ldots7}^t||_2 > \epsilon) \\
&\quad + G_{8,9}(\mathbf{s}_{8,9}^t)\mathbb{1}(||\mathbf{s}_{8,9}^t||_2 > \epsilon). \qquad (\epsilon\text{-NONSTATIONARYMP})
\end{aligned}
$$

**Spriteworld** Since we extended `Spriteworld` with a ground truth mask renderer, we are able to directly evaluate our SANDy models in `Spriteworld` as well. See the main text and Appendix B for a description of the `Spriteworld` environment.

### C.3 Results

In this Subsection we measure ability of the proposed SANDy algorithm (in its two variants) to correctly infer local factorization. At each transition we can query the environment for the ground-truth connectivity pattern of the local causal graph: given $|S| + |A|$ dimensions of current state and action and $|S|$ dimensions of next state, this corresponds an adjacency matrix $\mathbf{Y} \in \{0, 1\}^{|S|+|A| \times |S|}$. We note that accessing these evaluation labels—which are not used to train SANDy—requires a controlled synthetic environment like the ones we consider, and we leave design of an evaluation protocol suitable for real-world environments to future work.

We learn the SANDy network parameters using a training dataset of $40,000$ transitions, with an additional validation dataset of $10,000$ transitions used for early stopping and hyperparameter selection. We used the Adam optimizer with learning rate of $0.001$ and default hyperparameters. In the $\epsilon$-NONSTATIONARYMP, we set $\epsilon = 1.5$, while in the Spriteworld setting we collect training trajectories by deploying a random-action agent in the environment, and randomly resetting the environment with 5% probability at every step to increase diversity of experiences. We then evaluate the SANDy models by computing local factorizations $M_\tau(s, a) : |S| \times |A| \to \{0, 1\}^{(|S|+|A|) \times |S|}$ as a function of the threshold $\tau$ for each transition in a held-out test dataset of $10,000$ trajectories. We compute true and false positive rates for various values of $\tau$ to produce ROC plots.

Figure 9 shows that while the SANDy-Mixture is sufficient to solve the simpler synthetic MP settings (with avg AUC of $0.96$ and $0.91$ for the stationary and non-stationary variants), it scales poorly to the Spriteworld environment. While the modest inductive bias of sparse local connections and high-entropy mixture components in SANDy-Mixture makes it widely applicable, we hypothesize that its sensitivity to hyperparameters makes it difficult to tune in complex settings. Fortunately, SANDy-Transformer, performs favorably in Spriteworld by incorporating a stronger inductive bias about the state subspace structure. Thus we use SANDy-Transformer to perform local factorization inference in the remaining experiments.

Figure 10 provides some qualitative intuition as to how the two variants of SANDy differ in their attention strategy in the Spriteworld environment.

## D Fitting dynamics models to Spriteworld

Since CoDA is a data augmentation strategy, it is reasonable to consider an alternative approach to augmenting the experience buffer: sampling from a dynamics model as in model-based RL. Here we present some qualitative results from our efforts in fitting dynamics models to the Spriteworld environment. We found that while dynamics models achieve a decent error in the next-state prediction task, they fail to produce a diverse set of trajectories when used as autoregressive samplers. In particular, the autoregressive sampling did not model collisions well and often produced trajectories

(a) STATIONARYMP      (b) $\epsilon$-NONSTATIONARYMP      (c) Spriteworld

Figure 9: ROC plots for correct sparsity patter prediction on the three environments. On held-out test transitions we derive the ground truth local connectivity per step—label information that is *not* used to train the attention model—and measure (over 5 runs; 1 std. dev. shaded) true and false positive rates while sweeping the mask threshold $\tau$ over its allowable range. An accurate model generates an Area Under the Curve (AUC) close to 1. We observe that while SANDy-Mixture is sufficient for (nearly) solving the simpler synthetic MP environments, it underperforms in Spriteworld. SANDy-Transfomer, which has a stronger inductive bias, is sufficient to (nearly) solve Spriteworld.

where sprites converged to fixed points in space after a short number of steps. Figure 11 shows trajectories sampled autoregressively from Linear, MLP, and LSTM-based dynamics models, alongside the ground truth trajectory. Note that all dynamics models were trained to minimize error in next-state prediction given the current state and action. In other words the LSTM auto-regressively predicts successive dimensions of the next state rather than modeling multiple time steps of the trajectory. Nevertheless the environment is truly Markov because instantaneous velocities are observed, so this information should be sufficient in theory to capture the environment dynamics.

# E    Sample efficient dynamics modeling with CoDA

If we had access to the ground truth local factorization, even for a few samples (e.g., we could have humans label them), how much more efficient would it be to train a dynamics model? In Figure 12, we sample 2000 transitions from a random policy in Spriteworld and use the data to train a forward dynamics model using MSE loss. The validation loss throughout training is plotted. Our baseline uses only the initial dataset, and quickly overfits the training set, showing increasing error after the initial few epochs. The same applies to a "random" CoDA strategy, that does CoDA using an identity attention mask ($M(s,a) = I \ \forall \ (s,a)$) to randomly relabel the components. The random strategy does a bit better than the no CoDA strategy, since the randomness acts as a regularizer. Finally, we train a model using an additional 35,000 unique counterfactual CoDA transitions, and find that it significantly improves validation loss and prevents the model from overfitting. Note that we could have generated many more CoDA samples: from 2000 base transitions, if 80% of them do not involve collisions and there are 4 connected components in each, we could generate as many as $1600^4$ (6.5 trillion!) counterfactual samples.

# F    Compute Infrastructure

Experiments were run on a mix of local machines and a compute cluster, with a mix of GTX 1080 Ti, Titan XP, and Tesla P100 GPUs. This was solely to run jobs in parallel, and all experiments can be run locally (GPU optional for Spriteworld and Pong, but recommended for Fetch experiments).

(a) SANDy-Mixture　　　　　(b) SANDy-Transformer

Figure 10: Qualitative comparison of two attention mechanisms on the same `Spriteworld` trajectory. **SANDy-Mixture** (left) has a weaker inductive bias as it relies on sparsity regularization alone. Accordingly, it can learn a more compact subspace (e.g. grid patterns within a shape indicate that $x$ and $y$ coordinates move independently), but is less reliable in attending to collisions between shapes, and completely fails to attend to the action's affect on shapes. **SANDy-Transformer** has a stronger inductive bias and can more reliably infer the local interaction pattern between the five subspaces (four shapes and one action).

(a) Ground Truth

(b) Linear Dynamics

(c) MLP Dynamics

(d) LSTM Dynamics

Figure 11: Auto-regressive model-based rollouts for a variety of dynamics models fit to Spriteworld. While the dynamics models were able to achieve relatively low error in the next-state prediction task, they fail to capture collisions and long-term dependencies in the sampled trajectories, and thus were omitted as baselines in the RL experiments. All trajectories share the same initial state.

Figure 12: Learning curves for training a forward dynamics model using data from a random policy. Dotted lines indicate training performance, whereas solid lines indicate validation performance. We see that using the ground truth mask prevents overfitting and allows us to achieve much better validation performance.

[Supplementary Material 2 · coda_appendix.pdf]

# A  Proof of Proposition 1

**Lemma 1.** *If $V^j \in Pa^{\mathcal{L}}(V^i)$ in DAG $\mathcal{G}^{\mathcal{L}}$ of (local) causal model $\mathcal{M}^{\mathcal{L}}$, and $\mathcal{L} \subset \mathcal{X}$, then $V^j \in Pa^{\mathcal{X}}(V^i)$ in DAG $\mathcal{G}^{\mathcal{X}}$ corresponding to causal model $\mathcal{M}^{\mathcal{X}}$.*

*Proof.* By minimality, there exist $\{u^i, v^{-j}, v_1^j\}$ and $\{u^i, v^{-j}, v_2^j\}$ with $v^{-j} \in Pa^{\mathcal{L}}(V^i) \backslash V^j$ for which $f^i(\{u^i, v^{-j}, v_1^j\}) \neq f^i(\{u^i, v^{-j}, v_2^j\})$. Expand $\{v^{-j}, v_1^j\}$ and $\{v^{-j}, v_2^j\}$ to $(s_1, a_1), (s_2, a_2) \in \mathcal{L}$ (with any values of other components). But $\mathcal{L} \subset \mathcal{X}$, so $(s_1, a_1), (s_2, a_2) \in \mathcal{X}$ and it follows from minimality in $\mathcal{X}$ that $V^j \in Pa^{\mathcal{X}}(V^i)$. $\qquad\square$

**Corollary 1.** *If $\mathcal{L} \subset \mathcal{X}$, $\mathcal{G}^{\mathcal{L}}$ is sparser (has fewer edges) than $\mathcal{G}^{\mathcal{X}}$.*

**Proposition 1.** *The causal mechanisms represented by $\mathcal{G}_i, \mathcal{G}_j \subset \mathcal{G}$ are independent in $\mathcal{G}^{\mathcal{L}_1 \cup \mathcal{L}_2}$ if and only if $\mathcal{G}_i$ and $\mathcal{G}_j$ are independent in both $\mathcal{G}^{\mathcal{L}_1}$ and $\mathcal{G}^{\mathcal{L}_2}$, and $f^{\mathcal{L}_1, i} = f^{\mathcal{L}_2, i}, f^{\mathcal{L}_1, j} = f^{\mathcal{L}_2, j}$.*

*Proof.* ($\Rightarrow$) If $\mathcal{G}_i$ and $\mathcal{G}_j$ are independent in $\mathcal{G}^{\mathcal{L}_1 \cup \mathcal{L}_2}$, independence in $\mathcal{G}^{\mathcal{L}_1}$ and $\mathcal{G}^{\mathcal{L}_2}$ follows from Corollary 1. That $f^{\mathcal{L}_1, i} = f^{\mathcal{L}_2, i}$ (and $f^{\mathcal{L}_1, j} = f^{\mathcal{L}_2, j}$), on their shared domain, follows since each is a restriction of the same function $f^{\mathcal{L}_1 \cup \mathcal{L}_2, i}$ (or $f^{\mathcal{L}_1 \cup \mathcal{L}_2, j}$).

($\Leftarrow$) Suppose $\mathcal{G}_i$ and $\mathcal{G}_j$ are independent in $\mathcal{G}^{\mathcal{L}_1}$ and $\mathcal{G}^{\mathcal{L}_2}$ but not $\mathcal{G}^{\mathcal{L}_1 \cup \mathcal{L}_2}$. By the definition of independence applied to $\mathcal{G}^{\mathcal{L}_1 \cup \mathcal{L}_2}$, we have that, without loss of generality, there is a $V_i \in \mathcal{G}_i, V_j \in \mathcal{G}_j$ with $V_j \in Pa^{\mathcal{L}_1 \cup \mathcal{L}_2}(V_i)$. Then, from the definition of minimality, it follows that there exist $(s_1, a_1), (s_2, a_2) \in \mathcal{L}_1 \cup \mathcal{L}_2$ that differ only in the value of $V_j$, and $u_i \in \text{range}(U_i)$ for which $f^i(s_1, a_1, u_i) \neq f^i(s_2, a_2, u_i)$.

Clearly, if $(s_1, a_1)$ and $(s_2, a_2)$ are both in $\mathcal{L}_1$ (or $\mathcal{L}_2$), there will be an edge from $V_j$ to $V_i$ in $\mathcal{G}^{\mathcal{L}_1}$ (or $\mathcal{G}^{\mathcal{L}_2}$) and the claim follows by contradiction. Thus, the only interesting case is when, without loss of generality, $(s_1, a_1) \in \mathcal{L}_1$ and $(s_2, a_2) \in \mathcal{L}_2$. The key observation is that $(s_1, a_1)$ and $(s_2, a_2)$ differ only in the value of node $V_j \notin \mathcal{G}_i$: since $\mathcal{G}_i$ is an independent causal mechanism in both $\mathcal{G}^{\mathcal{L}_1}$ and $\mathcal{G}^{\mathcal{L}_2}$ and the parents of $V_i$ take on the same values in each, we have that $f^i(s_1, a_1, u_i) = f^{i, \mathcal{L}_1}(s_1, a_1, u_i) = f^{i, \mathcal{L}_2}(s_2, a_2, u_i) = f^i(s_2, a_2, u_i)$ and the claim follows by contradiction. $\qquad\square$

# B  Additional Experiment Details

This section provides training details for the experiments discussed in Section 4 as well as some additional results. Code is available at `https://github.com/spitis/mrl` [69].

## B.1  Online RL

Here we detail the procedure used in the Online RL experiments in the Spriteworld environment.

**Implementation**  We work with the original TD3 codebase, architecture, and hyperparameters (except batch size; see below), and focus our efforts solely on modifying the agent's training distribution.

**Environment**  We extend the base Spriteworld framework [89] as follows:

1. We add basic collisions, wherein a sprites velocity is reversed if it would overlap with another sprite on the next frame. This induces a local factorization and prevents a global factorization from being applicable.

2. We use a new continuous action space that consists of a 2-dimensional "click" on the canvas. The click changes the velocity of the closest sprite according to distance and angle between the click and the sprite's center.

3. We add a disentangled state renderer that returns the position and velocity of each sprite (a total of 16-dimensions in tasks with four sprites).

4. We add a mask renderer that returns the ground truth masking function. This allows us to evaluate our masking function in Appendix C.

5. We add a suite of partial and sparse reward tasks that we use for experiments. The tasks vary in difficulty along the following two dimensions

   - **Number of shapes:** The agent must place $N \in \{1, 2, 3, 4\}$ of the 4 sprites at their respective target (one of four quadrants).
   - **Sparsity of reward signal:** The agent receives reward either *Partial*, $\frac{1}{N}$ reward for each of the $N$ sprites placed near its target (up to a maximum reward of 1); or *Sparse*, reward of 1 is only observed when *all* $N$ sprites reach their target.

   For example, hardest task—"sparse reward, place 4" in the results—requires placing 4 shapes with sparse reward. Episodes run for 50 timesteps, and there is no termination signal (i.e., the environment is continuous from the agent's perspective). We also tried a variant with dense rewards, but found that the baseline agent was able to easily solve this task and no benefit was conferred by data augmentation.

These extensions will included with the release of our code.

**Data augmentation** Every 1000 environment steps, we sample 2000 pairs of random transitions from the agent's replay buffer, and apply CoDA to produce a maximum of 5 unique CoDA samples per pair. We apply two forms of CoDA, using (1) an oracle / ground truth mask function that we back out of the simulator, and (2) the mask of a pre-trained transformer model (see Section C). The mask function was trained using approximately 42,000 samples from a random policy (5/6 of 50,000, with the rest of the data used for validation). CoDA samples are added to a second CoDA replay buffer. For purposes of this experiment both buffers are have effectively infinite capacity (they are never filled). During training, the agent's batches are sampled proportionally from the real and CoDA replay buffers (this means that approximately 7/8 of the data that the agent trains on is counterfactual).

**Baselines** In addition to the base TD3 agent and CoDA, we also use the transformer model that is used as a mask function to generate data by performing forward rollouts with a random policy, as in Dyna [82]. So that this baseline produces approximately the same number of samples as CoDA with the learned mask, we roll the model out for 5 steps from 1500 random samples from the replay buffer, again every 1000 environment steps. This produces 7500 model-based samples for every 1000 environment transitions.

As an additional model-based Dyna baseline, we also fit a plain feedforward neural network with 2 hidden layers of 256 ReLU units as a feedforward model. We found that this forward model performed much worse than our transformer-based dyna baseline. The results are shown in Figure 6.

Figure 6: **Average reward in Partial reward setting with MLP-based Dyna baseline.**

Use of the transformer mask function for CoDA requires setting the threshold value $\tau$, which we do by monitoring accuracy and F1 scores for sparsity prediction (as discussed in Appendix C) on validation data, ultimately using the value $\tau = 0.05$.

**Batch Size** Since CoDA samples are plentiful we increase the agent's batch size from 256 to 1000 to allow it to train on more environment samples in the same number of training steps. We found that this slightly improved the performance of the base TD3 agent. An increase in batch size also allows the agent to see more of its own on-policy data in the face of many off-policy CoDA samples.

**Additional results in sparse reward task variants** In addition to the partial reward tasks described in Section 4, we also tested CoDA in four sparse reward tasks of varying difficulty. These are the same as the partial reward tasks, except that a sparse reward of 1 is granted only when *all* N sprites

Figure 7: **Average reward on Sparse reward bouncing ball environment.** As in the partial reward case (Figure 4), we observe that CoDA agents outperform the other agents (except in Place 4, where no agent achieves any reward).

are in their target locations. While these tasks were much harder (and perhaps impossible in the case of Place 4 due to moving sprites), as shown in Figure 7, the CoDA agents maintain a clear advantage.

**Results plot** The results plot shows the 3 seeds in reduced opacity together with their smoothed mean.

## B.2  Batch RL

Here we detail the procedure used in the Batch RL experiments in the `Pong` environment.

**Implementation** This experiment works with a different codebase than the Spriteworld experiment, in order to simplify the use of CoDA in a Batch RL setting. Our experiment first builds the agent's dataset (consisting of real data, dyna data and/or CoDA data), then instantiates a TD3 agent by filling its replay buffer with the dataset. The replay buffer is always expanded to include the entire enlarged dataset (for the 5x CoDA ratio at 250,000 data size this means the buffer has 1.5E6 experiences). The agent is run for 500,000 optimization steps.

**Hyperparameters** We used similar hyperparameters to the original TD3 codebase, with the following differences:

- We use a discount factor of $\gamma = 0.98$ instead of 0.99.
- Since Pong is a sparse reward task with $\gamma = 0.98$, we clip critic targets to $(-50, 50)$.
- We use networks of size $(128, 128)$ instead of $(256, 256)$.
- As for our Spriteworld experiment, we use a batch size of 1000.

**Environment** We base our `Pong` environment on `RoboSchoolPong-v1` [40]. The original environment allowed the ball to teleport back to the center after one of the players scored, offered a small dense reward signal for hitting the ball, and included a stray "timeout" feature in the agent's state representation. We fix the environment so that the ball does not teleport, and instead have the episode reset every 150 steps, and also 10 steps after either player scores. The environment is treated as continuous and never returns a done signal that is not also accompanied by a `TimeLimit.truncated` indicator [11]. We change the reward to be strictly sparse, with reward of $\pm 1$ given when the ball is behind one of the players' paddles. Finally, we drop the stray "timeout" feature, so that the state space is 12-dimensional, where each set of 4 dimensions is the x-position, y-position, x-velocity, and y-velocity of the corresponding object (2 paddles and one ball).

The opponent in the environment uses a small pretrained policy that was included in the original environment [40] (the effectiveness of this policy was unaffected by the removal of the timeout feature). To collect the batch dataset, we use the same policy as an "expert" policy, but replace 50% of its actions by random actions.

**Training the CoDA model** Without access to a ground truth mask, we needed to train a masking function C to identify local disentanglement. We also forewent the ground truth reward, instead training our own reward classifier. In each case we used the batch dataset given, and so we trained different models for each random seed. For our masking model, we stacked two single-head

transformer blocks (without positional encodings) and used the product of their attention masks as the mask. Each block consists of query $Q$, key $K$, and value $V$ networks that each have 3-layers of 256 neurons each, with the attention computed as usual [85, 47]. The transformer is trained to minimize the L2 error of the next state prediction given the current state and action as inputs. The input is disentangled, and so has shape (`batch_size, num_components, num_features`). In each row (component representation) of each sample, features corresponding to other components are set to zero. The transformer is trained for 2000 steps with a batch size of 256, Adam optimizer [38] with learning rate of 3e-4 and weight decay of 1e-5. For our reward function we use a fully-connected neural network with 1 hidden layer of 128 units. The reward network accepts an $(s, a, s')$ tuple as input (not disentangled) and outputs a softmax over the possible reward values of $[-1, 0, 1]$. It is trained for 2000 steps with a batch size of 512, Adam optimizer [38] with learning reate of 1e-3 and weight decay of 1e-4. All hyperparameters were rather arbitrary (we used the default setting, or in case it did not work, the first setting that gave reasonable results, as was determined by inspection). To ensure that our model and reward functions are trained appropriately (i.e., do not diverge) for each seed, we confirm that the average loss of the CoDA model is below 0.005 at the training and that the average loss of the reward model is below 0.1, which values were found by inspection of a prototype run. These conditions were met by all seeds.

When used to produce masks, we chose a threshold of $\tau = 0.02$ by inspection, which seemed to produce reasonable results. A more principled approach would do cross-validation on the available data, as we did for Spriteworld (Appendix C).

**Tested configurations**    Table 1 reports a subset of our tested configurations. We report our results in full below. We considered the following configurations:

1. **Real data only**.
2. **CoDA + real data**: after training the CoDA model, we expand the base dataset by either 2, 3, 4 or 6 times.
3. **Dyna (using CoDA model)**: after training the CoDA model, we use it as a forward dynamics model instead of for CoDA; we use 1-step rollouts with random actions from random states in the given dataset to expand the dataset by 2x. We also tried with 5-step rollouts, but found that this further hurt performance (not shown). Note that Dyna results use only 5 seeds.
4. **Dyna (using MBPO model)**: as the CoDA model exhibits significant model bias when used as a forward dynamics model, we replicate the state-of-the-art model-based architecture used by MBPO [34] and use it as a forward dynamics model for Dyna; we experimented with 1-step and 5-step rollouts with random actions from random states in the given dataset to expand the dataset by 2x. This time we found that the 5-step rollouts do better, which we attribute to the lower model bias together with the ability to create a more diverse dataset (1-step not shown). The MBPO model is described below. We use the same reward model as CoDA to relabel rewards for the MBPO model, which only predicts next state.
5. **MBPO + CoDA**: as MBPO improved performance over the baseline (real data only) at lower dataset sizes, we considered using MBPO together with CoDA. We use the base dataset to train the MBPO, CoDA, and reward models, as described above. We then use the MBPO model to expand the base dataset by 2x, as described above. We then use the CoDA model to expand *the expanded dataset* by 3x the original dataset size. Thus the final dataset is 5x as large as the original dataset (1 real : 1 MBPO : 3 CoDA).

All configurations alter only the training dataset, and the same agent architecture/hyperparameters (reported above) are used in each case.

**MBPO model**    Since using the CoDA model for Dyna harms rather than helps, we consider using a stronger, state-of-the-art model-based approach. In particular, we adopt the model used by Model-Based Policy Optimization [34]. This model consists of a size 7 ensemble of neural networks, each with 4 layers of 200 neurons. We use ReLU activations, Adam optimizer [38] with weight decay of 5e-5, and have each network output a the mean and (log) diagonal covariance of a multi-variate Gaussian. We train the networks with a maximum likelihood loss. To sample from the model, we choose an ensemble member uniformly at random and sample from its output distribution, as in [34].

**Full results**    See Table 2 for results.

| $|\mathcal{D}|$ (1000s) | Real data 1ʀ | Dyna 1ʀ:1ᴍ | MBPO 1ʀ:1ᴍ | Ratio of Real:CoDA [:MBPO] data (ours) | | | | |
|---|---|---|---|---|---|---|---|---|
| | | | | 1ʀ:1ᴄ | 1ʀ:2ᴄ | 1ʀ:3ᴄ | 1ʀ:5ᴄ | 1ʀ:3ᴄ:1ᴍ |
| 25 | $13.2 \pm 0.7$ | $12.2 \pm 1.4$ | $18.5 \pm 1.5$ | $43.8 \pm 2.7$ | $41.6 \pm 3.7$ | $40.9 \pm 2.5$ | $38.4 \pm 4.9$ | $\mathbf{46.8} \pm 3.2$ |
| 50 | $22.8 \pm 3.2$ | $17.4 \pm 3.0$ | $36.6 \pm 4.0$ | $66.6 \pm 3.7$ | $58.4 \pm 4.7$ | $64.4 \pm 3.1$ | $62.5 \pm 3.5$ | $\mathbf{70.4} \pm 3.7$ |
| 75 | $43.2 \pm 4.7$ | $23.3 \pm 4.7$ | $46.0 \pm 4.5$ | $73.4 \pm 2.8$ | $71.6 \pm 3.8$ | $\mathbf{76.7} \pm 2.6$ | $75.0 \pm 3.3$ | $74.6 \pm 3.3$ |
| 100 | $63.0 \pm 3.0$ | $25.9 \pm 7.4$ | $66.4 \pm 5.2$ | $77.8 \pm 2.0$ | $77.4 \pm 1.8$ | $\mathbf{82.7} \pm 1.5$ | $76.6 \pm 3.0$ | $73.7 \pm 2.9$ |
| 150 | $77.4 \pm 1.3$ | $34.1 \pm 1.5$ | $72.6 \pm 5.6$ | $82.2 \pm 1.7$ | $84.0 \pm 1.2$ | $\mathbf{85.8} \pm 1.4$ | $84.2 \pm 1.0$ | $79.7 \pm 3.5$ |
| 250 | $78.2 \pm 2.7$ | $44.2 \pm 4.0$ | $77.9 \pm 2.3$ | $85.0 \pm 2.8$ | $\mathbf{88.1} \pm 1.0$ | $87.8 \pm 1.7$ | $87.0 \pm 1.0$ | $78.3 \pm 5.0$ |

Table 2: Extended Batch RL results. Mean success ($\pm$ standard error, estimated using 1000 bootstrap resamples) on Pong environment. All results average over 10 seeds, except Dyna, which uses 5.

## B.3 Goal-conditioned RL

Here we detail the procedure used in the Goal-conditioned RL experiments on the Fetchpush-v1 and Slide2 environments.

**Implementation**  This experiment uses the same codebase as our Batch RL, which provides state-of-the-art baseline HER agents and will be released with the paper.

**Hyperparameters**  For Fetchpush-v1 we use the default hyperparameters from the codebase (see [70] for details on how they were selected), which outperform the original HER agents of [1, 71] and follow-up works. We do not tune the CoDA agent (but see additional CoDA hyperparameters below). They are as follows:

- Off-policy algorithm: DDPG [51]
- Hindsight relabeling strategy: futureactual_2_2 [68], using exclusively future [1] relabeling for the first 25,000 steps
- Optimizer: Adam [38] with default hyperparameters
- Batch size: 2000
- Optimization frequency: 1 optimization step every 2 environment steps after the 5000th environment step
- Target network updates: update every 10 optimization steps with a Polyak averaging coefficient of 0.05
- Discount factor: 0.98
- Action l2 regularization: 0.01
- Networks: 3x512 layer-normalized [4] hidden layers with ReLU activations
- Target clipping: (-50, 0)
- Action noise: 0.1 Gaussian noise
- Epsilon exploration [71]: 0.2, with an initial 100% exploration period of 10,000 steps
- Observation normalization: yes
- Buffer size: 1M

On Slide2 we tried to tune the baseline hyperparameters somewhat, but note that this is a fairly long experiment (10M timesteps) and so only a few settings were tested due to constraints. In particular, we considered the following modifications:

- Expanding the replay buffer to 2M (effective)
- Reducing the batch size to 1000 (effective)* (used for results)
- Using the future_4 strategy (agent fails to learn in 10M steps)
- Reducing optimization step frequency to 1 step every 4 environment steps (about the same performance)

We tried similar adjustments to our CoDA agent, but found the default hyperparameters (used for results) performed well. We found that the CoDA agent outperforms the base HER agent on all tested settings.

For CoDA, we used the following additional hyperparameters:

- CoDA buffer size: 3M
- Make CoDA data every: 250 environment steps
- Number of source pairs from replay buffer used to make CoDA data: 2000
- Number of CoDA samples per source pair: 2
- Maximum ratio of CoDA:Real data to train on: 3:1

**Environment**   On `FetchPush-v1` the standard state features include the relative position of the object and gripper, which entangles the two. While this could be dealt with by dynamic relabeling (as used for HER's reward), we simply drop the corresponding features from the state.

Figure 8: The `Slide2` environment.

`Slide2` has two pucks that slide on a table and bounce off of a solid railing. Observations are 40-dimensional (including the 6-dimensional goal), and actions are 4-dimensional. Initial positions and goal positions are sampled randomly on the table. During training, the agent gets a sparse reward of 0 (otherwise -1) if *both* pucks are within 5cm of their ordered target. At test time we count success as having both picks within 7.5cm of the target on the last step of the episode. Episodes last 75 steps and there is no done signal (this is intended as a continuous task).

**CoDA Heuristic**   For these experiments we use a hand-coded heuristic designed with domain knowledge. In particular, we assert that the action is always entangled with the gripper, and that gripper/action and objects (pucks or blocks) are disentangled whenever they are more than 10cm apart. This encodes independence due to physical separation, which we hypothesize is a very generally heuristic that humans implicitly rely on all the time. The pucks have a radius of 2.5cm and height of 4cm, and the blocks are 5cm x 5cm x 5cm, so this heuristic is quite generous / suboptimal. Despite being suboptimal, it demonstrates the ease with which domain knowledge can be injected via the CoDA mask: we need only a high precision (low false positive rate) heuristic—the recall is not as important. It is likely that an agent could learn a better mask that also takes into account velocity.

**Results plot**   The plot shows mean $\pm 1$ standard deviation of the smoothed data over 5 seeds.

## C   Inferring Local Factorization

Here we present several approaches to inferring the local factorization of subspaces, a crucial subroutine of CoDA. We note that in many cases where domain knowledge is available, simple heuristics may suffice, e.g. in our Goal-conditioned RL experiments discussed in Section 4 where a simple distance-based indicator function in the state space was used. However, as such heuristics may not be universally available, the question of whether data-driven approaches can successfully infer local factorization is of general interest. We note that the performance of CoDA will improve alongside future improvements in this inference task (motivating future work in this area), and that inferring the local factorization in general is an easier task than learning the environment dynamics.

We begin by presenting two methods for inferring local factorization, derived from variants of a next-state prediction task, which we here refer to as SANDy for Sparse Attention Neural Dynamics. To verify the merits of SANDy in the local factorization inference task, we evaluate two SANDy variants in controlled settings where the ground truth factorization is known: first in a synthetic Markov process (MDP without actions), and second in Spriteworld. This label information is used only to evaluate performance, and not to train the SANDy parameters. The SANDy-Transformer model was used for the Online RL experiments presented in Section 4.

## C.1 Methods

We propose two Sparse Attention for Neural Dynamics (SANDy) models. In each case we seek to learn a function (or mask) $M(s, a) \rightarrow \{0, 1\}^{(|S|+|A|) \times |S|}$ whose output represents the adjacency matrix of the local causal graph, conditioned on the state and action. We note that $M(s, a)_{i,j} = 0$ alone is insufficient, in general, to determine the local subspace $\mathcal{L} \subset \mathcal{S} \times \mathcal{A}$, since there may be multiple disconnected subspaces $\mathcal{L}_k$ with $M_k(s, a)_{i,j} = 0$ whose union $\bigcup_k \mathcal{L}_k$ has $M_{\cup k}(s, a)_{i,j} = 1$. This can be resolved by our Proposition 1 if we also force the relevant structural equations to be the same. For now, we assume the mask determines the local subspace, and leave exploration of this possibility to future work. Empirically, we will see that our assumption is reasonable.

**SANDy-Mixture** The first model is a mixture-of-MLPs model with an attention mechanism that is computed from the current state. Each component of the mixture is a neural dynamics model with sparse local dependencies. For a given component, the key idea is to train a neural dynamics model to predict the next state $h(s_t, a_t) \approx s_{t+1}$ and approximate the masking function by thresholding the (transpose of the) network Jacobian of $h$, $[\mathbf{J}(s, a)]_{i,j} = \frac{\delta}{\delta[s,a]_j}[h(s, a)]_i$. Intuitively, we can think of $\mathbf{J}$ as providing the first-order element-wise dependencies between the predicted next state and the network input. We then derive the local factorization by thresholding the absolute Jacobian

$$M_\tau(s, a) = \mathbb{1}(|\mathbf{J}(s, a)| > \tau), \tag{1}$$

where $\mathbb{1}(\cdot)$ represents the indicator function and $\tau$ is a threshold hyperparameter.

To estimate the network Jacobian, we note that for standard activation functions (sigmoid, tanh, relu), it can be bound from above by the matrix product of its weight matrices. To see this, let $h_\theta$ be an $L$-layer MLP parameterized by $\theta = (\mathbf{W}^{(1)}, \mathbf{b}^{(1)}, \cdots, \mathbf{W}^{(L)}, \mathbf{b}^{(L)})$ with activation $\sigma$ with bounded derivative $\sigma'(x) \leq 1$, and note that for each layer $\boldsymbol{h}^{(\ell)} = \sigma(\mathbf{W}^{(\ell)}\boldsymbol{h}^{(\ell-1)} + \mathbf{b}^{(\ell)})$ we have:

$$\left| \frac{d\boldsymbol{h}_j^{(\ell)}}{d\boldsymbol{h}_i^{(\ell-1)}} \right| \leq \left| \mathbf{W}_{ji}^{(\ell)} \right|.$$

Then, using the chain rule, triangle inequality, and the identity $|ab| = |a||b|$, we can compute:

$$\left| \frac{d\boldsymbol{h}_j^{(\ell)}}{d\boldsymbol{h}_i^{(\ell-2)}} \right| \leq \left| \frac{d}{d\boldsymbol{h}_i^{(\ell-2)}} \mathbf{W}_{j\cdot}^{(\ell)} \boldsymbol{h}^{(\ell-1)} \right| = \left| \mathbf{W}_{j\cdot}^{(\ell)} \cdot \frac{d\boldsymbol{h}^{(\ell-1)}}{d\boldsymbol{h}_i^{(\ell-2)}} \right|$$

$$\leq \sum_k \left| \mathbf{W}_{jk}^{(\ell)} \frac{d\boldsymbol{h}_k^{(\ell-1)}}{d\boldsymbol{h}_i^{(\ell-2)}} \right| \leq \left| \mathbf{W}_{j\cdot}^{(\ell)} \right| \cdot \left| \mathbf{W}_{\cdot i}^{(\ell-1)} \right|.$$

Expanding this out to multiple layers, we see that $|\mathbf{J}_\theta(s, a)| \leq \prod_{\ell \in [L]} |\mathbf{W}^{(\ell)}|$, as desired. We use this upper bound to approximate the network Jacobian of an MLP by setting $\hat{\mathbf{J}} = \prod_{\ell \in [L]} |\mathbf{W}^{(\ell)}|$. A similar idea is used by [25, 44], among others, to control element-wise input-output dependencies.

We use this static approximation $\hat{\mathbf{J}}$ to facilitate learning a sparse dynamic mask by specifying a mixture model $h(s, a) = \sum_i \alpha_\phi^{(i)}(s, a)h_\theta^{(i)}(s, a)$ over the environment dynamics (with $\sum_i \alpha_\phi^{(i)}(s, a) = 1 \ \forall \ (s, a)$), where each component $h_\theta^{(i)}$ is an MLP as specified above with a sparsity prior on its Jacobian bound $\hat{\mathbf{J}}^{(i)}$ to encourage sparse (i.e. well-factorized) local solutions. The dynamic mask is computed by first approximating the Jacobian, $\hat{\mathbf{J}}(s, a) = \sum_i \alpha_\phi^{(i)}(s, a)\hat{\mathbf{J}}^{(i)}$, then thresholding by $\tau$ as in Equation 1.

Note that we assume the mixture components $\alpha$ are a function of the current state and action alone; in other words the factorization (captured by the network Jacobian of each component) can be *locally inferred*. The next-state prediction is given by $\hat{\mathbf{s}}_{t+1} = (h_\theta^{(1)}(\mathbf{s}_t, \mathbf{a}_t), \cdots, h_\theta^{(N)}(\mathbf{s}_t, \mathbf{a}_t))^T \boldsymbol{\alpha}_\phi(\mathbf{s}_t, \mathbf{a}_t)$. To train the model, we optimize the objective:

$$\underset{\theta, \phi}{\text{minimize}} \ \frac{1}{|\mathcal{D}|} \left( \sum_{(\mathbf{s}_t, \mathbf{a}_t \mathbf{s}_{t+1}) \in \mathcal{D}} ||\mathbf{s}_{t+1} - h(\mathbf{s}_t, \mathbf{a}_t)||_2^2 \right) + \lambda_1 S(\theta) + \lambda_2 R(\phi) + \lambda_3 ||(\theta, \phi)||_2 \tag{2}$$

where $S(\theta) = \frac{1}{K}\sum_i |\hat{\mathbf{J}}_\theta^{(i)}|_1$ puts an $\ell_1$ prior on each mixture component to induce sparsity, and $R(\phi)$ encourages high entropy in the attention probabilities

$$R(\phi) = \frac{1}{|\mathcal{D}|}\sum_{\mathbf{s}\in\mathcal{D}}\sqrt{\frac{1}{N}\sum_{j\in[N]}[A_\phi(\mathbf{s},\mathbf{a})]_j}.$$

We note that more sophisticated methods of computing Jacobians of neural networks–including architectural changes and optimization strategies [3]—have been proposed, and could in principle be used here as well.

**SANDy-Transformer**   As an alternative model, we use a transformer-like architecture that applies self-attention between a set of (potentially multi-dimensional) inputs [85, 47]. Our architecture is composed of a stacked self-attention blocks. Each block accepts a set of inputs $X = \{x_i \in \mathbb{R}^n\}$ and composes three functions of each input: query $Q : \mathbb{R}^n \to \mathbb{R}^d$, key $K : \mathbb{R}^n \to \mathbb{R}^d$, and value $V : \mathbb{R}^n \to \mathbb{R}^m$. The block returns a set of outputs $\{y_i \in \mathbb{R}^m\}$ of size $|X|$, each of which is computed as: $y_i = A_i^T V(X)$, where $A_i = \text{softmax}\left(\sum_j (Q(x_i)_j K(x_j)_i)\right)$ (note that $V(X)$ is a matrix of size $|X| \times m$). We approximate the block mask (approximate Jacobian) as $A = [A_1, A_2, \ldots, A_{|X|}] \in \mathbb{R}^{|X|\times|X|}$, and the full network mask as the product of the block masks (as in the SANDy-Mixture). We used two-layer MLPs for each function $K, Q, V$ in Spriteworld and three-layer MLPs in Pong.

To apply this architecture to our problem, we first embed each state and action component (single feature, or group of features) into $\mathbb{R}^n$ to produce a set of inputs $X$ and pass this through each stacked self-attention block to obtain a set of outputs $Y$. We then discard any output components that correspond to the action features to obtain the next state prediction and mask. The network $h$ is trained to minimize mean squared error:

$$\underset{\theta}{\text{minimize}}\ \frac{1}{|\mathcal{D}|}\sum_{(\mathbf{s}_t, \mathbf{a}_t \mathbf{s}_{t+1})\in\mathcal{D}} ||\mathbf{s}_{t+1} - h(\mathbf{s}_t, \mathbf{a}_t)||_2^2. \tag{3}$$

Unlike the SANDy-Mixture, no sparsity regularizers are applied, as we found the sparsity induced by the softmax attention mechanism to be sufficient.

## C.2   Evaluation environments

**Synthetic Markov Processes**   We investigate the capacity of the SANDy-Mixture to learn simple factorized transition dynamics under a globally factored Markov Process (STATIONARYMP). Unlike the general MDP setting, no agent/policy is considered. However the ability to train an unconstrained dynamics model to approximate factorized environment dynamics is an important subtask within our overall approach. Assuming a spherical Gaussian prior over the initial states $\mathbf{s}^0 \in \mathbb{R}^9$, the STATIONARYMP is entirely specified by transition distribution $p(\mathbf{s}^{t+1}|\mathbf{s}^t)$. We assume that the state transitions factorize (globally) into three parts. Denoting by $\mathbf{s}_{n\cdots m}^t$ the $n$-th through $m$-th dimensions of the time $t$ state $\mathbf{s}^t$, we have:

$$p(\mathbf{s}_{1\ldots9}^{t+1}|\mathbf{s}_{1\ldots9}^t) = p(\mathbf{s}_{1\ldots4}^{t+1}|\mathbf{s}_{1\ldots4}^t)p(\mathbf{s}_{5\ldots7}^{t+1}|\mathbf{s}_{5\ldots7}^t)p(\mathbf{s}_{8,9}^{t+1}|\mathbf{s}_{8,9}^t).$$

In other words, we have a block-diagonal transition matrix comprising three blocks with sizes 4, 3, and 2, respectively. In our case, all transition factors are deterministic non-linear mappings, e.g. $p(\mathbf{s}_{1\ldots4}^{t+1}|\mathbf{s}_{1\ldots4}^t) = \delta(g_{1\ldots4}(\mathbf{s}_{1\ldots4}^t))$, with $g_{1\ldots4} : \mathbb{R}^4 \to \mathbb{R}^4$ is a randomly-initialized single-hidden-layer neural network with 32 hidden units and GELU nonlinearity [30]. In this case, we can alternatively express the deterministic dynamics via the transition function

$$\begin{aligned}
\mathbf{s}^{t+1} &= (\mathbf{s}_{1\ldots4}^{t+1}, \mathbf{s}_{5\ldots7}^{t+1}, \mathbf{s}_{8,9}^{t+1}) \\
&= (g_{1\ldots4}(\mathbf{s}_{1\ldots4}^t), g_{5\ldots7}(\mathbf{s}_{5\ldots7}^t), g_{8,9}(\mathbf{s}_{8,9}^t)). \tag{STATIONARYMP}
\end{aligned}$$

We now turn to a more sophisticated MP with locally factored dynamics ($\epsilon$-NONSTATIONARYMP), and investigate whether the SANDy-Mixture can learn to recognize local disentanglement.

We refer to the component functions $g_{1\ldots4} : \mathbb{R}^4 \to \mathbb{R}^4$, $g_{5\ldots7} : \mathbb{R}^3 \to \mathbb{R}^3$, and $g_{8,9} : \mathbb{R}^2 \to \mathbb{R}^2$ used in the STATIONARYMP as *local* transitions. We now introduce *global* interactions via the global transition functions, $G_{1\ldots4} : \mathbb{R}^4 \to \mathbb{R}^9$, $G_{5\ldots7} : \mathbb{R}^4 \to \mathbb{R}^9$, and $G_{8,9} : \mathbb{R}^4 \to \mathbb{R}^9$, which respectively

map the local state factors onto the global state space[7]. This allows us to extend the STATIONARYMP by adding global state transitions to the local state transitions whenever the norm of the local state factors exceeds the value of a hyperparameter $\epsilon$ (lower $\epsilon$ indicates more global interaction). Denoting by $\mathbb{1}(\cdot)$ the indicator function, we have

$$
\begin{aligned}
\mathbf{s}^{t+1} &= \left[ (\mathbf{s}_1^{t+1}, \mathbf{s}_2^{t+1}, \mathbf{s}_3^{t+1}, \mathbf{s}_4^{t+1}), (\mathbf{s}_5^{t+1}, \mathbf{s}_6^{t+1}, \mathbf{s}_7^{t+1}), (\mathbf{s}_8^{t+1}, \mathbf{s}_9^{t+1}) \right] \\
&= \left[ g_{1\ldots4}(\mathbf{s}_{1\ldots4}^t), g_{5\ldots7}(\mathbf{s}_{5\ldots7}^t), g_{8,9}(\mathbf{s}_{8,9}^t) \right] \\
&\quad + G_{1\ldots4}(\mathbf{s}_{1\ldots4}^t)\mathbb{1}(||\mathbf{s}_{1\ldots4}^t||_2 > \epsilon) \\
&\quad + G_{5\ldots7}(\mathbf{s}_{5\ldots7}^t)\mathbb{1}(||\mathbf{s}_{5\ldots7}^t||_2 > \epsilon) \\
&\quad + G_{8,9}(\mathbf{s}_{8,9}^t)\mathbb{1}(||\mathbf{s}_{8,9}^t||_2 > \epsilon). \qquad (\epsilon\text{-NONSTATIONARYMP})
\end{aligned}
$$

**Spriteworld**   Since we extended `Spriteworld` with a ground truth mask renderer, we are able to directly evaluate our SANDy models in `Spriteworld` as well. See the main text and Appendix B for a description of the `Spriteworld` environment.

### C.3   Results

In this Subsection we measure ability of the proposed SANDy algorithm (in its two variants) to correctly infer local factorization. At each transition we can query the environment for the ground-truth connectivity pattern of the local causal graph: given $|S| + |A|$ dimensions of current state and action and $|S|$ dimensions of next state, this corresponds an adjacency matrix $\mathbf{Y} \in \{0,1\}^{|S|+|A| \times |S|}$. We note that accessing these evaluation labels—which are not used to train SANDy—requires a controlled synthetic environment like the ones we consider, and we leave design of an evaluation protocol suitable for real-world environments to future work.

We learn the SANDy network parameters using a training dataset of $40,000$ transitions, with an additional validation dataset of $10,000$ transitions used for early stopping and hyperparameter selection. We used the Adam optimizer with learning rate of $0.001$ and default hyperparameters. In the $\epsilon$-NONSTATIONARYMP, we set $\epsilon = 1.5$, while in the Spriteworld setting we collect training trajectories by deploying a random-action agent in the environment, and randomly resetting the environment with 5% probability at every step to increase diversity of experiences. We then evaluate the SANDy models by computing local factorizations $M_\tau(s, a) : |S| \times |A| \to \{0,1\}^{(|S|+|A|) \times |S|}$ as a function of the threshold $\tau$ for each transition in a held-out test dataset of $10,000$ trajectories. We compute true and false positive rates for various values of $\tau$ to produce ROC plots.

Figure 9 shows that while the SANDy-Mixture is sufficient to solve the simpler synthetic MP settings (with avg AUC of $0.96$ and $0.91$ for the stationary and non-stationary variants), it scales poorly to the Spriteworld environment. While the modest inductive bias of sparse local connections and high-entropy mixture components in SANDy-Mixture makes it widely applicable, we hypothesize that its sensitivity to hyperparameters makes it difficult to tune in complex settings. Fortunately, SANDy-Transformer, performs favorably in Spriteworld by incorporating a stronger inductive bias about the state subspace structure. Thus we use SANDy-Transformer to perform local factorization inference in the remaining experiments.

Figure 10 provides some qualitative intuition as to how the two variants of SANDy differ in their attention strategy in the Spriteworld environment.

## D   Fitting dynamics models to Spriteworld

Since CoDA is a data augmentation strategy, it is reasonable to consider an alternative approach to augmenting the experience buffer: sampling from a dynamics model as in model-based RL. Here we present some qualitative results from our efforts in fitting dynamics models to the Spriteworld environment. We found that while dynamics models achieve a decent error in the next-state prediction task, they fail to produce a diverse set of trajectories when used as autoregressive samplers. In particular, the autoregressive sampling did not model collisions well and often produced trajectories

|  (a) STATIONARYMP | (b) $\epsilon$-NONSTATIONARYMP | (c) `Spriteworld` |

Figure 9: ROC plots for correct sparsity patter prediction on the three environments. On held-out test transitions we derive the ground truth local connectivity per step—label information that is *not* used to train the attention model—and measure (over 5 runs; 1 std. dev. shaded) true and false positive rates while sweeping the mask threshold $\tau$ over its allowable range. An accurate model generates an Area Under the Curve (AUC) close to 1. We observe that while SANDy-Mixture is sufficient for (nearly) solving the simpler synthetic MP environments, it underperforms in Spriteworld. SANDy-Transfomer, which has a stronger inductive bias, is sufficient to (nearly) solve Spriteworld.

where sprites converged to fixed points in space after a short number of steps. Figure 11 shows trajectories sampled autoregressively from Linear, MLP, and LSTM-based dynamics models, alongside the ground truth trajectory. Note that all dynamics models were trained to minimize error in next-state prediction given the current state and action. In other words the LSTM auto-regressively predicts successive dimensions of the next state rather than modeling multiple time steps of the trajectory. Nevertheless the environment is truly Markov because instantaneous velocities are observed, so this information should be sufficient in theory to capture the environment dynamics.

## E  Sample efficient dynamics modeling with CoDA

If we had access to the ground truth local factorization, even for a few samples (e.g., we could have humans label them), how much more efficient would it be to train a dynamics model? In Figure 12, we sample 2000 transitions from a random policy in Spriteworld and use the data to train a forward dynamics model using MSE loss. The validation loss throughout training is plotted. Our baseline uses only the initial dataset, and quickly overfits the training set, showing increasing error after the initial few epochs. The same applies to a "random" CoDA strategy, that does CoDA using an identity attention mask ($M(s, a) = I \ \forall \ (s, a)$) to randomly relabel the components. The random strategy does a bit better than the no CoDA strategy, since the randomness acts as a regularizer. Finally, we train a model using an additional 35,000 unique counterfactual CoDA transitions, and find that it significantly improves validation loss and prevents the model from overfitting. Note that we could have generated many more CoDA samples: from 2000 base transitions, if 80% of them do not involve collisions and there are 4 connected components in each, we could generate as many as $1600^4$ (6.5 trillion!) counterfactual samples.

## F  Compute Infrastructure

Experiments were run on a mix of local machines and a compute cluster, with a mix of GTX 1080 Ti, Titan XP, and Tesla P100 GPUs. This was solely to run jobs in parallel, and all experiments can be run locally (GPU optional for `Spriteworld` and `Pong`, but recommended for `Fetch` experiments).

(a) SANDy-Mixture         (b) SANDy-Transformer

Figure 10: Qualitative comparison of two attention mechanisms on the same `Spriteworld` trajectory. **SANDy-Mixture** (left) has a weaker inductive bias as it relies on sparsity regularization alone. Accordingly, it can learn a more compact subspace (e.g. grid patterns within a shape indicate that $x$ and $y$ coordinates move independently), but is less reliable in attending to collisions between shapes, and completely fails to attend to the action's affect on shapes. **SANDy-Transformer** has a stronger inductive bias and can more reliably infer the local interaction pattern between the five subspaces (four shapes and one action).

(a) Ground Truth

(b) Linear Dynamics

(c) MLP Dynamics

(d) LSTM Dynamics

Figure 11: Auto-regressive model-based rollouts for a variety of dynamics models fit to Spriteworld. While the dynamics models were able to achieve relatively low error in the next-state prediction task, they fail to capture collisions and long-term dependencies in the sampled trajectories, and thus were omitted as baselines in the RL experiments. All trajectories share the same initial state.

Figure 12: Learning curves for training a forward dynamics model using data from a random policy. Dotted lines indicate training performance, whereas solid lines indicate validation performance. We see that using the ground truth mask prevents overfitting and allows us to achieve much better validation performance.

## Footnotes

[7] Like the local transition functions, global transition functions are implemented as randomly-initialized single-hidden-layer neural networks.