[Reviews · NeurIPS 2020]

Review 1

Summary and Contributions: In this paper the authors propose Counterfactual Data Augmentation (CoDA), a technique for generating unseen sample trajectories to improve the efficiency of Reinforcement Learning (RL) algorithms. Taking a causal graphical view of the MDP, the authors take advantage of regions of the action-state space in which additional independences hold relative to the global graph of the transition model, producing simplified "local causal models" (LCMs). From the LCMs, samples associated with independent components can be effectively mixed and matched to produce new counterfactual samples. They also show how a number of existing data augmentation methods can be cast as particular versions of CoDA. Experiments across a number of RL settings (online, batch, and goal-conditioned) show that CoDA can significantly improve sample complexity. ----------------- UPDATE post-rebuttal: Thanks to the authors for their clarifications in the rebuttal. I continue to believe the contributions of this paper are valuable so I am maintaining my original score. During discussion with the other reviewers, it became clear that the manuscript would really benefit from some more attention to detail and improved clarity regarding certain aspects of the method. In particular, making sure the minimality condition is correctly and precisely defined, and adding more discussion about the role of the causal discovery/graph learning piece of CoDA. I hope the authors will make good use of the revision to make these improvements---I think the resulting paper will make a great contribution to NeurIPS.

Strengths: I enjoyed reading this paper. LCMs are an interesting idea and were very clearly explained. Further, given the mask, CoDA seems straightforward to implement and generally applicable. The experimental results were strong and promising. In particular, even when the mask was learned there were appreciable gains in performance.

Weaknesses: In general, I thought this paper was quite strong. The main questions I had were: 1) how often do we expect the locally factorable dynamics to arise? and 2) how difficult is it to learn the local factorization (and how important is being correct)? Regarding 1: The authors motivate the procedure with examples (e.g., the two-armed robots) in which often the "sub-processes" operate independently, but I was left wondering *how much* simpler the LCM needs to be to see advantages. As the number of components increases, the amount of counterfactual data that can be generated should increase. But what if an LCM only contains 1 component? Is this something that can happen, and can counterfactual data still be generated? Regarding 2: What happens when the learned local factorization is misspecified? In the Spriteworld experiments CoDA (learned) still provides improvements over the baseline, but could a misspecified mask ever hurt performance? It would be nice to see an experiment in which the gap between CoDA (ground truth) and CoDA (learned) is plotted as a function of some measure of the degree to which the learned mask is misspecified. As the authors state in the appendix, inferring the local factorization is a key part of CoDA, so to the extent possible it would be very helpful for the paper to discuss or study the sensitivity of the procedure to this piece.

Correctness: To the best of my knowledge the results are correct.

Clarity: This paper was very well-written. Section 2, in particular, was easy to follow and set the stage for the rest of the paper. Figures 2&3 were very helpful for getting a quick understanding of the proposed method and how it works.

Relation to Prior Work: The discussion of related methods within the body were well done. For example, the description of existing data augmentation techniques as specific instances of CoDA was nicely explained. The Related Work section itself, by comparison, was somewhat lacking. Given more space in the revisions, I think the authors could briefly expand on how the different concepts that are discussed relate to the current paper (e.g. , Factored MDPs, object-oriented approaches to RL, and policy-gradient baselines).

Reproducibility: Yes

Additional Feedback:


Review 2

Summary and Contributions: This paper proposes an approach to causally-motivated reinforcement learning by considering local relationships in the underlying decision process. Doing so, the authors argue, permits more efficient estimation in both offline and online settings.

Strengths: In general, this paper is very much not in line with my expertise. What follows is my best effort to provide feedback with respect to the RL contributions (viewed as strengths by default due to my unfamiliarity with the corresponding literature). Weaknesses below focus mostly on the causal aspects and issues of clarity, as this is what I am best suited to judge (and therefore am able to find issues that need clarification or possibly correction). The paper is well written. The motivation throughout the introduction and continuing into the main body of the paper is cohesive and well-paced. The examples provided, billiards and the two-armed robot, are intuitive and provide a solid grounding for why the proposed approach might be useful. Theoretical Grounding: The authors provide a rather detailed background on the formalisms underlying their approach. This does not necessarily extend to formal analysis of the approach itself. Empirical Evaluation: I am only marginally familiar with the subtle details of the various toy problems (e.g. Atari) commonly studied in the RL literature. That said, the approach to evaluation that the authors give seems sound. Significance and novelty: From a causal inference perspective, this work does seem interesting and novel. I find some of the details regarding sample complexity hazy and I question how the authors can obtain an increase in effective sample size in the way they describe. I seek clarification on this point below and aside from this concern the work seems to provide a promising avenue for improved ability to perform policy evaluation.

Weaknesses: Theoretical: The authors provide little formal justification for their approach. One of the main contributions seems to be the increase of effective sample size by performing data augmentation. What is unclear is why the increase actually happens. In remark 3.1, the authors attempt to answer the question "How much data can we generate using model-free CoDA?", claiming an exponential increase in data. This fact is not immediately clear. In an attempt to frame this analysis in terms of the more familiar (to this reviewer) causal inference setting, consider the following: we have data collected from a sequential decision-making process that is representable by a causal DAG, potentially with some Markovian structure (i.e. to preclude the possibility that the graph is complete), where the final variable in the graph is the outcome of interest Y. Then, the analyst/agent wishes to evaluate how intervening on some decision variable in the graph A by setting A according to a function (policy) affects Y. In counterfactual notation, the goal is to estimate E[Y(fA)]. Based on causal identification theory (see e.g. papers by Jin Tian's 2008 paper on defining dynamic treatment regimes and Shpitser's 2018 UAI paper which provides completeness results for Tian's paper, as well as various papers by Elias Barenboim and papers that consider estimating these effects with semiparametric methods (broadly, see the work of Robins, Van der Laan, Diaz, Kennedy, etc.)), this quantity is expressable (under the no-confounding type assumption the authors make) as a function of the variables in the model, which itself can be broken up into several univariate conditional distributions which must be estimated from the available data to get the estimate of the counterfactual. This learning process -- of the univariate conditionals -- means that the result is effectively equivalent to density estimation (though in practice this is usually avoided in CI). One can certainly then sample from these estimated densities but I do not see how this helps the overall estimation procedure. There is no _new_ information encoded in these new samples, even if clever factorizations are possible because of local independence among the variables in the model. Perhaps I am grossly misinterpreting the author's argument here but if the paper is meant to bridge the gap between the RL and causal literatures, this point should be made more clear. Additionally, the authors should provide a formal argument in the way of a proposition and proof, rather than arguing/stating informally in the remark given. Aside from this issue, it is not totally clear when the subspaces described at the bottom of page 3 exist. What might the causal graph look like that enables this local factorization to be performed? If the authors are invoking causal models as in section 2.1 then a graphical criterion should be derivable. In line with the above requests for clarification, the authors' minimality assumption seems somewhat imprecise. They say "intuitively, minimality says that Vj is a parent of Vi if and only if setting the value of Vj can have an effect on the child Vi through the structural equation fi". Take the graph V1 -> V2 -> V3. Intervening on V1 will have an (indirect) effect on V3, which is propagated through the structural equation fV2 by recursive substitution. That is: V3 <- f_V3(f_V2(f_V1(epsilon_V1), epsilon_V2), epsilon_V3) and so setting the value of V1 will have an effect on V3. This violates the iff relationship since V1 is not V3's parent. In the next two paragraphs the authors suggest (paraphrasing) that such relationships are generally very dense in the graph but that they are frequently not substantive and so attention can be restricted to subspaces where the relationships are not substantive for the sake of more efficient learning. Is this the correct reading of this point? Given the above apparent issue with the definition of parent/child relationships, is this subspace argument still valid? It would seem that the reframing necessary induces a very dense graph. Without a clear understanding of the frequency of the independence described on line 108, it is not obvious that there are many cases where the necessary subspaces can be identified/isolated.

Correctness: See above comments in the weaknesses section. I cannot properly verify the RL-related claims given my relative lack of expertise on the relevant literature and methods.

Clarity: For the most part, yes. See the above comments on the related causal concepts.

Relation to Prior Work: The paper cites very thoroughly. With respect to the RL literature I cannot judge the degree of novelty. With respect to the causal inference literature, I am not familiar with any work that pursues this sort of methodological contribution. The authors may, however, consider citing the papers mentioned above about estimating the effects of policy interventions ("dynamic treatment regimes) in causal inference for completeness.

Reproducibility: Yes

Additional Feedback: UPDATE AFTER DISCUSSION The authors, (or perhaps more accurately the other reviewers), adequately cleared up my confusions about this paper. I have increased my score to a weak accept, maintaining the confidence score. I do want to emphasize that as the authors consider implementing revisions for a subsequent draft, they should take care to clear up definitions and discussions of minimality. Additionally, if the authors wish for this paper to be read by a causal audience, they should draw clearer connections with the causal literature by making sure to frame their contributions in a causal lens (e.g. discussing Markov properties with respect to "shallowness" -- mentioning the graphs are "shallow" in the first place might also be a good idea).


Review 3

Summary and Contributions: The authors formally introduce local causal models (LCM), that partition a global structural causal model into local models that only apply in a subspace of the whole state space. Within one LCM and its respective subspace, state dimensions can be independent that may otherwise interact. This fact can be exploited by mixing the independent parts of transition samples to generate unseen (counterfactual) data and is formalized as counterfactual data augmentation. Besides the theoretical formalization, the authors provide an algorithm to discover local causal dependencies given a disentangled state space, and show how this can improve the sample efficiency of RL-agents in different tasks. They demonstrate that their approach outperforms two existing data augmentation techniques, namely HER and Dyna.

Strengths: I like the counterfactual causal framing of the data augmentation strategy, it is a valid way of looking at this kind of data augmentation through a causal lense and the improvement in using the newly generated data is (mostly) clear. The approach is in itself novel, to the best of my knowledge. The method is elegant in the sense that it is not complicated to implement (modulo the causal discovery part), but provides significant benefits.

Weaknesses: The approach assumes that the state and action space is nicely disentangled/interpretable in order to use the benefits of a discrete DAG structure. The counterfactual data augmentation depends heavily on the learned causal graph, yet in the main text of the paper the discussion about finding the causal graph is minimal and is mostly shifted to the appendix. The method for causal discovery has no guarantee of finding the causal graph (looking at the thresholded Jacobian and using a transformer). The counterexamples are not stated and assumptions not clearly argued for which cases this method would or would not work. The design choice of using a Transformer-based mask is not as well motivated and seems a bit ad hoc. The theoretical formalization is sometimes not precise enough and at times feels confusing (subgraphs vs. mechanisms?, structural equations vs. mechanisms?).

Correctness: I have to focus on the claim of using causal graphs with the causal discovery method proposed in the appendix. Except for the methods being ad-hoc, they also seem wrong, there is no guarantee that the DAG is causal and under which assumptions, this is not clearly argued in the main text. (see additional feedback) Some experiments are not explained enough. (see additional feedback)

Clarity: It is (relatively) simple to grasp the general idea of the paper, but certain parts are difficult to digest. In the section concerning the implementation and the section on the experiments more details are needed to fully understand the paper. (see additional feedback)

Relation to Prior Work: This work connects well to the literature on data augmentation and causal RL. In particular, the authors do a good job in relating and contrasting existing data augmentation approaches, such as pixel-based image augmentation or HER, to their approach.

Reproducibility: No

Additional Feedback: 10: The algorithm is described as 'model-free' which confused me, since the discovery of a local causal model is required for this approach (without full domain knowledge). My understanding is that no model is needed for simulating the augmented data, which is why the approach is considered as 'model-free', correct? This could be elaborated more in the main text. 40: 'Figure 1' not linked 58: Typo: Either 'The basic model for decision making in a controlled dynamic process ...' or 'The basic model for decision making in controlled dynamic processes ...' 163: again, I am not sure if it is clear that this method is model-free, needs to be explained more. 183-184: I do not understand the last sentence of this paragraph and this 'example' confused me more than it actually helped understand the remark. 189: A definition or proper description of 'connected components' would be good in this context. Is it correct that this refers to one subgraph? I would suggest adding a proper definition of 'connected components' here, earlier at line 150, or avoid using the term 'connected components' alltogether. Algorithm 1, function CoDA: \tilde{a}[d] is replaced by s2[d]. Shouldn't this be a2[d]? Algorithm 1, function Components: Why are the columns of the action replaced by dummy colums? Please elaborate on this. Figure 2: why are your noise variables not independent? Figure 3: please state in the caption what do the full vs dashed lines denote and also the coloring of the nodes in the graphs. Table 1: why doesn’t more CoDA samples always increase performance? Should be commented. 197: “require causal mechanisms for … to share structural equations”, are not causal mechanisms structural equations? Intermixing the term subgraph with mechanism is also confusing. 211: The use of a Transformer and its attention mask for deriving M seems out of the blue here. A motivation or intuition for this should be added. Why does the attention mask capture the causal dependencies required by M? What is the minimum requirement for extracting M? Can other attention-based architectures be applied for this as well? 225: The Spriteworld environment from [83] seems to be heavily modified. However, the paper lacks a proper explanation of this scenario. What do the two-dimensional actions represent, seeing that previously the action space was four-dimensional? Do the objects move on their own, and if so, how do they move? Please add more details on the scenario or describing the modifications- otherwise the task at hand cannot be understood. 234: Here the transformer architecture is used "for fair comparison". However, I don't think this is the case, since the two approaches (CoDa and Dyna) use completely different components of the network: CoDA uses the internal mask, while Dyna relies on the output prediction. I think it would greatly improve the paper, if several model-based approaches were compared to CoDA. In the appendix it says that other models were omitted as baselines since they fail "to capture collisions and long-term dependencies in the sampled trajectories". The images suggest, that the networks converge to some identity function to minimize the prediction error. A common practice for this would be predict state changes (\Delta s) instead of next states, since this does not allow the trivial solution of the identity function. How does this affect the model-based approach? Besides that, why wasn't the MBPO used as a baseline in Experiment 1, as well? 254: How are the results gathered in the batch-RL task? From pong plays against an opponent, since this is what according to [35] the Pong scenario is for? If so, what type of opponent was used? Regarding your method to causal discovery, say that I give you variables A, B, C and the causal graph A->B, A->C. C doesn’t cause B but it can strongly correlated with B. How would your method to finding the DAG deal with this situation and therefore how would the counterfactual data augmentation work? There is a clear need for more discussion about this in the main paper, since causal formalism is heavily used and currently this part is missing. Again, more discussion and motivation about the causal discovery method is needed in the main text in my opinion. Otherwise, I like the approach and I think that it’s relevant. --update-- I have increased my score since the authors addressed more-or-less all of the relevant critique points. I still think there should be a bit more discussion on the discovery method in the main text, the causal formalism could be connected better to the method, although I realize that it is orthogonal work.


Review 4

Summary and Contributions: This work is focused on the powerful observation that even if on the global scale the dynamics model is densely connected (each pair of subprocesses do eventually interact at some point), locally the structure is sparser and the dynamic processes can be factorized in causally independent subprocesses. First, the paper introduces the Local Causal Model (LCM), a strictly sparser SCM generated from the global model by conditioning on a subspace of states. Second, a Counterfactual Data Augmentation (CoDA) method is proposed that makes counterfactual modification to local causal factors by mixing locally independent subsamples of the environment in order to create new samples that are causally valid. In experiments, CoDA augmentation shows improved sample efficiency and superior results in locally factorized environments from the online, batch-RL and goal-conditioned settings.

Strengths: - The sparser LCM is a general tool, suitable for modelling a wide range of dynamic processes in a factorized way. I believe that this local decomposition could be used beyond the counterfactual augmentation goal, to increase the efficiency and performance in different areas that suffer from densely connected global graphs, but that can be locally sparsified as done in this work. - The proposed algorithm for augmentation is simple and intuitive while being nicely connected and motivated by theory. - The experimental part is carefully designed and shows superior results in all three scenarios. The gap in results between the ground truth matrix and the generated mask promises even more improvement in results if a better model will be used to generate the local causal graph and also if the constraints imposed by Proposition 1 will be more closely followed.

Weaknesses: The paper points out an attractive idea about the locally sparse causal model and applied it in an interesting way to augment the states of the environment. While I agree that the current experimental settings more clearly emphasize the concepts presented in the paper, it would be great to see how valid CoDA augmentation is when working on pixel-level and if the improvement is still preserved for noisier graphs. Beside this, I enjoyed the paper and I don’t see any major weakness.

Correctness: The model proposed for local dynamics is sound and the algorithm follows the theoretical findings. All the misalignments between them are discussed in the paper.

Clarity: The paper is well written and self-explanatory. All the details necessary to reproduce the work are included in the supplementary material, together with the proof for the theoretical remark.

Relation to Prior Work: The paper mentions all the relevant work that I am aware of, and nicely refers to them both in the related work and during the paper.

Reproducibility: Yes

Additional Feedback: Minor comments: - In Fig. 3 d) I think there is a typo in the second graph (S_{i+1} instead of S_{j+1}). - In Supplementary material, in (row 736) it is not clear what the asterix should mean. ======================== UPDATE =============================== I read the author's feedback and other reviewers' opinions. As someone that is not very familiar with this paper’s domain, I find this work well written and the idea presented simple and valuable for RL community. Thus, I maintain my original score, but I emphasize that there are elements regarding the causal inference theory that I am not very familiar with.

[Author Response · NeurIPS 2020]

**[R2, R3] Amount of augmented data and sample efficiency** R2 questioned the effective sample size of CoDA, and
R3 asked why more CoDA samples don't always increase performance. In response, we emphasize the distinction
between (1) the number of raw data samples and (2) the usefulness of those samples, and discuss some of the difficulties
associated with making precise statements about the latter. The exponential increase in *raw data samples* that results
from permuting different subspaces (our Remark 3.1) is plain: given m sets (subspaces), each with n unique elements
(subsamples), their Cartesian product has cardinality $n^m$. E.g. Permuting triplets $\{A1\clubsuit, B2\blacklozenge, C3\spadesuit, D4\heartsuit\}$ forms
$4^3 = 64$ unique samples: $(\{A1\clubsuit, A1\blacklozenge, A1\spadesuit, A1\heartsuit, A2\clubsuit, \dots\})$. This is all we meant by Remark 3.1: that within
a local subspace, we might produce an exponential number of raw data samples (if there are duplicate subsamples,
the permuted samples will contain duplicates). This does not necessarily mean that sample efficiency increases by a
commensurate amount (we observe in the Batch RL experiment that too much CoDA data can hurt). Several reasons
prevent us from making a stronger statement than Remark 3.1. First is distributional shift, too much of which can cause
off-policy algorithms to diverge (the same effect is observed with HER [1], Fig 6). Unfortunately we are not aware of
any good theoretical approach to quantifying the effect of distributional shift in RL (cf. [17] and arxiv:2003.07305 ).
Second is the fact that CoDA is model-agnostic: it occurs at the data level, and can be applied to models with different
inductive biases. We agree with R2: if the model already has the local SCM built-in as an inductive bias, and learns by
estimating each of the univariate conditionals as posited, CoDA will not help. Such a model is not generally available
in the RL context, and in our paper we feed CoDA data into standard agents that use fully connected networks with
no built-in notion of independence / subspaces. In this case, CoDA data prevents the model from learning spurious
dependencies across independent subspaces, and helps with sample efficiency, as we demonstrate empirically. We note
that to the extent that non-causal dependencies (between features in the state at time $t$) are useful for the prediction task
(policy or value), especially in the low-data regime, CoDA data might help the model learn these non-causal relations;
thus, we believe CoDA will be helpful even when using a factored model such as a GNN (perhaps fruitful future work).

**[R2] "Minimality" and Project Scope** We agree our "intuitive" explanation of minimality might mislead in the way
described, and will append something to the effect of "while holding all other nodes constant" to better reflect the formal
definition at L89-91, which we think is precise/precludes R2's counterexample. Note that causal graphs in our paper
are shallow (bipartite), so the $V_1 \rightarrow V_2 \rightarrow V_3$ example would not arise in our context. Overall, rather than unifying
RL/Causal literatures, we show a broad application of causal techniques yielding empirical sample efficiency in RL.

**[R1, R3] Causal discovery method** R3 notes that our choice of Transformer is a bit ad hoc (we agree) and that there is
no guarantee of finding the causal graph (we agree). We restricted our discussion of causal discovery to the appendix
because it is orthogonal to our main contributions [LCMs, Alg 1, empirical (see L44-55)] and is, as R3 correctly points
out, somewhat ad hoc in that alternatives are not as well explored as we would have liked. We note, however, that the
use of a network mask (thresholded absolute Jacobian) for (global) causal discovery is a known, theoretically-grounded
technique (see GraN-DAG [39], cited at L217), and that all we are doing is additionally conditioning the mask on the
local context $(s_t, a_t)$. This might done with any context-dependent model (e.g., the mixture of experts model we tried
in the Appdx), but we chose a Transformer because it was the first model we tried that worked well. As we note at
L290, this component can likely be improved in future work (as evidenced by the gap between learned CoDA and
ground truth CoDA in Fig. 4). We will add the figure R1 requested to the paper for the Batch RL experiment, but we
unfortunately could not complete it in time for the rebuttal. We note that a random mask fails (see Fig 11 in Appdx), and
have observed anecdotally that using a poorly trained Transformer model for CoDA results in poor agent performance.

**[R1, R3] When do local factorizations (subspaces at bottom of page 3) arise?** We see that our examples at the
end of Remark 3.4 are poorly worded and will revise. We expect that local factorizations will arise when things
are physically separate, as we observe in our various examples (two-armed robot, pool balls, sprites, pong paddles,
gripper and block), which we posit is very common (detection of physical separation should also transfer well across
different objects, as demonstrated in our Fetch experiments). The "mental ignorance" comment at the end of Remark
3.4 references a common scenario in multi-agent settings, where agent A's actions are known to be independent of agent
B's actual thoughts (but not agent A's belief about agent B's thoughts), and other true facts that agent A is ignorant of.

**[R3] Model-based baseline.** MBPO baseline was omitted for Spriteworld only because the experiment used a different
codebase (see submitted code), and we only implemented the stronger MBPO model in the Batch RL code. Anecdotally,
we tested a couple models (e.g, LSTM) in Spriteworld, but they performed poorly; we will add a standard (non-
Transformer) model in the revision. We did not try the delta state trick; this is a helpful suggestion (thanks!) that we
believe will help the learned CoDA mask as well. Note that CoDA + MBPO were complementary in the Batch RL case.

**[R3] Precision (subgraphs/mechanisms, model-free, reproducibility)** Thanks for these comments. We will do our
best to clarify in the revision. For now, we note that (1) mechanism and subgraph are more or less interchangeable (see
definition L148); neither is meant to include the structural equation, (2) we think our usage of model-free (defined at
L66) is standard in RL, insofar as we reuse subsamples from the environment rather than rolling out a forward dynamics
model, and (3) code was provided for reproducibility (but that is no excuse for missing details, which we will add).

[Meta-Review · NeurIPS 2020]

Reviewers were positive and excited about the paper, and I agree with the general sentiment that the work is a significant step in the right direction. My recommendation is "accept." Having said that, there are some issues that I would like to see fixed to make its final version more comfortable to read, sound, consistent, and well-positioned regarding the broader literature. Towards this goal, first, read the reviews carefully and try to incorporate their feedback as much as you can. I will list some critical issues below, mostly in addition to the ones raised by the reviewers. — The definition of minimality is not consistent and may lead to problems in other parts of the paper (as discussed in the reviews). Please, re-define causal model to account for the bipartite structure mentioned in the rebuttal; that's a strong constraint over the SCM-space but appears to be enough for the paper's purposes. That's a serious issue and shouldn't be overlooked. — It's a common source of confusion the discussion on model-based versus model-free, which is about having or not the specific parameterization of the underlying model (e.g., P(S | S', A), P(R | S, A)). This paper assumes much more than a parametric model of the dynamics, namely, detailed knowledge of the causal structure itself. While this is okay, to claim that the proposed procedure is "model-free" is somewhat far-fetched. — The idea of mixing and matching independent parts of the data, coming from distinct mechanisms, seems quite nice. The notion of context-specific independence (CSI, see Boutilier, Koller, et al.) seems to be the key behind the current approach. Contribution #3 for using "attention-based" methods & "disentangle state space" seems a bit like a distraction (even though a nice one). Instead of just "augmenting data," why not have an algorithm that considers the corresponding CSIs? This is a less desired mode of learning (e.g., people rotate/scale images, and feed into convolution NNs because CNN can't handle 'rotation' in general. ) For instance, in the robot's example with left, right arms, one can further use the symmetry relationship where 'left arm' data is fed into 'right arm' with mirroring. Data augmentation is all about the data generating process and prior knowledge, not counterfactuals. — Data based on CoDA might be selection biased or causally-invalid. Samples to be mixed satisfy some kind of "independence criteria", and conditioning on such criteria will result in selection bias. The authors seem to acknowledge, very briefly, such a selection bias is possible in Remark 3.3. Selection bias can adversely affect the agent's performance, but the paper does not perform any experiments demonstrating robustness against this bias. Furthermore, imbalanced data highlighting augmented samples without 'interaction' may harm agents' performance for the cases where subprocesses interact. Readers may appreciate if you add some acknowledgment of this phenomenon from an SCM-graphical perspective; for example, see discussion in Bareinboim and Pearl, 2012 2016 for more details on the semantics and available conditions needed for recoverability of data from selection bias. Further, consider a scenario with two initial conditions C1 and C2, where each results in (transition1A, transition1B) and (transition2A, transition2B). Assume that they can be safely mixed according to CoDA so that we can get new samples (transition1A, transition2B) and (transition2A, transition1B). However, it might be the case, under the condition C1, (transition1A, transition2B) is impossible. Imagine the billiard setting and let "transition1A" being one ball moves along the top side, and "transition2B" being another ball moves along the bottom side (in parallel). In a friction-free environment, it seems impossible and not "causally-valid". There must be a serious discussion on the meaning of "causally-valid" data with respect to the plausibility and selection-bias. — The notion of counterfactual is about events that cannot be realized in the real world (see Pearl, Ch. 7), where no data is available. The use of this expression seems inconsistent with the literature since data is factual, about one specific, realized world. Since some causal readers may find this usage somewhat off-putting, I would recommend adding a footnote explaining the difference from the interpretation employed in the paper.